# Occurrence and stability of hetero-hexamer associations formed by β-carboxysome CcmK shell components

**Luis F. Garcia-Alles**[1]*, **Katharina Root**[2], **Laurent Maveyraud**[3], **Nathalie Aubry**[1], **Eric Lesniewska**[4], **Lionel Mourey**[3], **Renato Zenobi**[2], **Gilles Truan**[1]

**1** Toulouse Biotechnology Institute (TBI), Université de Toulouse, CNRS, INRA, INSA, Toulouse, France, **2** Department of Chemistry and Applied Biosciences, ETH Zurich, Zurich, Switzerland, **3** Institut de Pharmacologie et Biologie Structurale, IPBS, Université de Toulouse, CNRS, UPS, Toulouse, France, **4** ICB UMR CNRS 6303, University of Bourgogne Franche-Comte, Dijon, France

* lgarciaa@insa-toulouse.fr

**Data Availability Statement:** All relevant data are within the manuscript and its Supporting Information files. Data pertaining the crystal

## Abstract

The carboxysome is a bacterial micro-compartment (BMC) subtype that encapsulates enzymatic activities necessary for carbon fixation. Carboxysome shells are composed of a relatively complex cocktail of proteins, their precise number and identity being species dependent. Shell components can be classified in two structural families, the most abundant class associating as hexamers (BMC-H) that are supposed to be major players for regulating shell permeability. Up to recently, these proteins were proposed to associate as homo-oligomers. Genomic data, however, demonstrated the existence of paralogs coding for multiple shell subunits. Here, we studied cross-association compatibilities among BMC-H CcmK proteins of *Synechocystis sp*. PCC6803. Co-expression in *Escherichia coli* proved a consistent formation of hetero-hexamers combining CcmK1 and CcmK2 or, remarkably, CcmK3 and CcmK4 subunits. Unlike CcmK1/K2 hetero-hexamers, the stoichiometry of incorporation of CcmK3 in associations with CcmK4 was low. Cross-interactions implicating other combinations were weak, highlighting a structural segregation of the two groups that could relate to gene organization. Sequence analysis and structural models permitted the localization of interactions that would favor formation of CcmK3/K4 hetero-hexamers. The crystallization of these CcmK3/K4 associations conducted to the elucidation of a structure corresponding to the CcmK4 homo-hexamer. Yet, subunit exchange could not be demonstrated *in vitro*. Biophysical measurements showed that hetero-hexamers are thermally less stable than homo-hexamers, and impeded in forming larger assemblies. These novel findings are discussed within the context of reported data to propose a functional scenario in which minor CcmK3/K4 incorporation in shells would introduce sufficient local disorder as to allow shell remodeling necessary to adapt rapidly to environmental changes.

structure is available from the RCBS protein databank, with accession code 6SCR.

**Funding:** LFGA, LM, NA, EL, LM and GT work received no specific funding for this work. KR and RZ acknowledge financial support for mass spectrometry studies by the Swiss National Science Foundation (grant number 200020_159929). The funders had no role in study design, data collection and analysis, decision to publish, or preparation of the manuscript.

**Competing interests:** The authors have declared that no competing interests exist.

# Introduction

Bacterial microcompartments (BMC) are protein-based organelles that encapsulate enzymes participating in a given metabolic route [1, 2]. An ever-increasing number of types of BMC that sustain varied catalyzed processes are being found in bacteria. Indeed, analysis of genomic data revealed that BMC likely form in a widespread manner across bacterial phyla [3]. BMC likely promote their specific processes by concentrating the enzymes and substrates in a limited volume, and also by sequestering toxic or volatile reaction intermediates. These properties, their natural diversity, the possibility to reprogram BMC contents by means of targeting peptides [4–6], the modularity evidenced by the fact that bricks from different BMC could be assembled together [7, 8], as well as the possibility to reconstitute BMC in recombinant hosts by operon transfer [9–12], justify the strong interest for BMC as prototypes for engineering future nano-reactors for synthetic biology purposes.

An intense effort has been devoted to the structural characterization of BMC [1]. Information for individual components was largely obtained by means of X-ray crystallography. Thus, high resolution structures for several dozen shell subunits from different BMC types are now available. These data, combined with sequence information, confirmed that whilst differing in enzymatic contents, all BMC shells are built from homologous proteins adopting two structural folds. The first one (Pfam00936) is present in proteins that are organized as hexamers (BMC-H) [13–18], and as tandem domains in subunits that form trimers (BMC-T) [18–20]. These proteins often give rise within crystals to stacked two-dimensional (2D) layers, a property that was also confirmed by electron microscopy and atomic force microscopy [21–23]. Consequently, these proteins were proposed early to compose the planar facets of BMC shells, which according to various electron microscopy studies, could be polyhedral/icosahedral [24–26]. The second type of domain (Pfam03319) normally associates as pentamers (BMC-P) [18, 27, 28], although an hexamer was described as part of a structural genomics program [29]. BMC-P were speculated to occupy pentagonal vertices of icosahedral shells. Overall, these basic principles were confirmed recently by the 3.5-Å resolution 3D structure of the first full BMC [30]. This impressive work also demonstrated the structural plasticity for shell components. Thus, two different hexamer-hexamer interactions, planar or 30°-tilted, were established in the single BMC-H subunit from *Haliangium ochraceum*, giving credit to the physiological meaning of non-planar assemblies described for BMC-H [23, 31, 32].

High-resolution structural investigations of shell components have been invariably performed on single proteins, which associate as homo-oligomers. However, it is now well established that variable number of genes coding for homologous BMC-H, BMC-T or BMC-P subunits are present in every BMC-carrying organism. Kerfeld and colleagues demonstrated that each genome contains, on average, 3.5 BMC-H genes, 1.4 BMC-T genes, and 1.2 BMC-P genes, with extreme cases having up to 15 BMC-H, 5 BMC-T or 7 BMC-P copies [3]. Moreover, several paralogs are often integrated within the same operon, and therefore are likely expressed simultaneously. It is therefore necessary to ascertain whether hetero-oligomers might form or not, since subunit properties such as pore size, pore and surface electrostatics, oligomer symmetry, or even assembly behavior might be considerably altered, when compared to homo-oligomers.

Here, we investigated this possibility by selecting the four CcmK paralogs that compose the shells of carboxysomes from the *Synechocystis sp.* PCC 6803 cyanobacteria (*Syn6803*). In this microorganism, *ccmK1* and *ccmK2* genes (coding for BMC-H paralogs) are encoded in an operon that also contains ccmL (BMC-P) and genes for other compartment components, whereas *ccmK3* and *ccmK4* cluster together in a different locus (a schematic representation of gene organization is presented in reference [33]). CcmK capacity to associate within hexamers

was inspected here by co-expressing protein couples in *E. coli*, and under configurations permitting simultaneous production of the four paralogs. Experimental compatibilities between CcmK1 and CcmK2, as well as between CcmK3 and CcmK4 paralogs could be demonstrated. The formation of hetero-hexamers between CcmK3 and CcmK4 from *Synechococcus elongatus* PCC7942 (*Syn7942*) could be shown too, thus confirming and complementing results obtained on the same proteins and on paralogs from *Halothece* PCC 7418 (*Hal7418*) [34]. Such observations were reasoned on the basis of sequence identities and compensatory effects for residues implicated in inter-subunit contacts, also by means of dynamic simulations using homology models. The stability of purified hetero-associations was finally investigated by biophysical means and in subunit exchange experiments. Overall, our data together with recent findings strongly suggest that the formation of such hetero-oligomers is likely to occur in cyanobacteria. Although adding to the complexity of BMC shells, this phenomenon might play important roles in modifying the structural robustness and environmental adaptability of BMC shells.

## Materials and methods

### Cloning

Full-length DNA sequences coding for shell proteins from *Synechocystis sp*. PCC6803, as well as for *ccmK3* and *ccmK4* from *Synechococcus elongatus* PCC7942 were synthesized (Genecust and Twist Bioscience, provided in S1 List). Sequences included stretches for N-ter or C-ter tag extensions, exception made of untagged proteins. To characterize individual proteins, sequences were cloned in pET15b using XbaI/XhoI sites. For co-expression studies, manipulations were carried out in pBlueScript II SK+ (Stratagene) after cloning between SacI and KpnI sites a synthetic sequence that comprised four cassettes with independent T7 promoter/lac operator, RBS and T7 terminators (shown in S1 List). For co-expression of protein couples, the fourth and second cassette were removed stepwise, using AvrII or BamHI/AgeI restriction enzymes (blunt ends prepared by reaction with Klenow fragment LC, Thermofisher), respectively, followed by plasmid recircularization after each step. The different ccmK sequences were integrated using SwaI/BamHI (1st cassette), PacI/AgeI (2nd), MfeI/SalI (3rd) or BsrGI/HindIII (4th), respectively. All sequences are detailed in S1 List. Treatments with BglII/BlpI permitted the transfer of final products to pET-26b. Resulting vectors were used to transform chemically-competent BL21(DE3) *E. coli* cells, following standard protocols.

### Expression, solubility and protein purification

All cell cultures corresponding to combinations with a given His-tagged protein were carried out in parallel, applying strictly the same protocol. Handled typical volumes were often 10 to 30 mL. When the cultures growing in LB at 37˚C reached mid log phase ($OD^{600nm}$ = 0.6–0.8), expression was induced with 0.2 mM IPTG (final conc.). Incubation was continued for 3–4 hours before cells were harvested at 6000 g and supernatant (SN) discarded. In studies of expression of 4 combined proteins, in order to increase yields of purified material, experiments were also carried out in ZYM-5052 auto-induction media [35]. After inoculation with $1/100^{th}$ volume of an overnight saturated pre-culture in LB, incubations were shaken at 220 rpm for 15 hours, at 37˚C.

Cellular lysis was carried out in $1/10^{th}$ of culture volume of 20 mM NaPi, 300 mM NaCl, 10 mM imidazole, pH 8), supplemented with DNase I (5 μg/mL final conc.) and lysozyme (0.05 mg/mL). Protease inhibitors aprotinin (10 μM, final conc.), leupeptin (20 μM) and pepstatin (2 μM) were also present. After incubation at room temperature with gentle agitation for 5 to 10 minutes, cells were sonicated at 4˚C. Four cycles of 30 sec sonication at 25% power, spaced by 1 min lags were applied (SO-VCX130 equipped with a 630–0422 probe, Sonics). The

inhibitor phenylmethylsulfonyl fluoride was added right after the first cycle (PMSF, 1 mM). Insoluble debris were removed by centrifugation for 20 min at 20000 x g (4°C). The supernatant (soluble fraction) was applied to cobalt-loaded TALON Superflow metal affinity resin (Clontech) preconditioned in Sol A (20 mM NaPi, 300 mM NaCl, 10 mM imidazole, pH 8.0). After thoroughly washing with Sol A, elution was performed with Sol B (300 mM imidazole in Sol A). A single fraction was collected, to which EDTA (5 mM final conc.) was added immediately after elution. SDS–polyacrylamide gels (15%) for Coomassie staining or Western Blots were run using Tris-Glycine-SDS buffer after loading heat denatured samples (95°C, 10 min, in loading dye (LD) preparation for SDS-PAGE) prepared from freshly lysed cells, from soluble fractions and from purified material. Loaded volumes of lysed cells and purified fractions were identical for all samples corresponding to combinations with a given His-tagged protein. In the case of soluble fractions, volumes were adjusted taking into consideration absorption values at 280 nm of supernatants.

When required, proteins were buffer-exchanged against Sol C (10 mM HEPES, 300 mM NaCl, pH 7.5) by 3–4 10-fold dilution/concentration steps in Vivaspin Turbo 15, 10 kDa MWCO devices, before being concentrated to 1–2 mg/mL. Protein concentrations were estimated from 280 nm absorption readings, using theoretical extinction coefficients calculated from protein sequences by the ExPASy ProtParam tool (http://web.expasy.org/protparam/).

## Western blot analysis

After SDS-PAGE, gel contents were electro-transferred to a PVDF membrane (Immobilion-P, Milipore). Membranes were blocked at room temperature (rt) for 1 hr in 5% nonfat dry milk in TBS containing 0.05% Tween 20. Standard protocols were applied for subsequent treatments. Primary antibody immunolabelling was carried out for 1hr at rt with 1:2000 diluted FLAG tag mouse monoclonal antibody (FG4R, ThermoFisher). After incubation with the secondary alkaline phosphatase-conjugated goat Anti-Mouse IgG (H+L) secondary antibody, and extensive washings, blots were developed with the Sigmafast BCIP/NBT substrate.

## Native mass spectrometry

Prior to analysis, protein tags were removed by incubation overnight at 25°C with turbo TEV protease (GenWay, 5 μg/mL final) in 50 mM Tris pH 8.0 / 300 mM NaCl /2 mM DTT/ 1 mM EDTA. After exchanging buffer by Sol C, proteins were concentrated to 1–2 mg/mL. Right prior to spraying, the samples were buffer-exchanged against 150 mM aqueous ammonium acetate at pH 8 using Amicon Ultra-0.5 mL centrifugal filters (MWCO = 10 kDa; Millipore). MS measurements were performed in positive ion mode exactly as described before [23]. For tandem mass spectrometry experiments, precursor ions were isolated in the quadrupole mass analyzer and accelerated into an argon-filled linear hexapole collision cell (P = 3.0 x $10^{-2}$ mbar). Various collision energy offsets were applied upstream of the collision cell.

## Size-exclusion chromatography

Protein sizes were estimated by SEC using a Beckman Ultraspherogel SEC2000 column (7.5 x 300 mm) mounted on a Waters 2690 HPLC separation module. Samples (10–20 μL) were injected at 1 mL/min flowrate after conditioning the column in 20 mM Tris-HCl, 300 mM NaCl at pH 7. Elution was monitored with a Waters 996 Photodiode Array Detector. Elution volumes (280 nm absorption) were used to estimate protein MW by comparison to calibration standards run under identical conditions: Ferritin (440 kDa), Aldolase (158 kDa), Conalbumin (75 kDa), Ovalbumin (43 kDa) and Ribonuclease (13.7 kDa).

## CcmK3 and CcmK3/K4 homology models and molecular dynamics simulations

Homology models for CcmK3 *Syn6803* homo-hexamers were built using SWISS-MODEL and PHYRE2 algorithms. Templates selected for 3D model reconstruction differed between the two, corresponding to the *Syn7942* CcmK1/2 (PDB ID 4OX7, residue identity of 53%) using SWISS-MODEL, or the structure of *Syn6803* CcmK4 (2A10, 47% identity) with PHYRE2. Hetero-hexamers were recomposed by replacing one of the monomers of the 2A10 CcmK4 structure by CcmK3 modeled monomers (previously superimposed by minimizing RMSD of monomer main-chain atoms). Few side-chain clashes between monomers in recomposed hexamers were relaxed using steepest descent energy minimization approaches.

Molecular dynamics simulations were carried using the AMBER14 forcefield implemented within YASARA software. After a first energy minimization within YASARA, hexamers were hydrated within a cubic cell with dimensions extending 10 Å beyond edge protein atoms, which was filled with explicit solvent. Periodic boundary conditions were applied. YASARA's pKa utility was used to assign residue protonation states at pH 7.0. The simulation cell was neutralized with NaCl (0.9% (w/v) final concentration) by iteratively placing sodium and chlorine ions at the coordinates with the lowest electrostatic potential. The cut-off for the Lennard-Jones potential and the short-range electrostatics was 8 Å. Long-range electrostatics were calculated using the Particle Mesh Ewald (PME) method with a grid spacing of 1.0 Å, $4^{th}$ order PME-spline, and PME tolerance of $10^{-5}$ for the direct space sum. The entire system was energy-minimized using steepest descent minimization, in order to remove conformational stress, followed by a simulated annealing minimization until convergence ($<0.05$ kJ/mol/200 steps). Simulations were run at 298 K, with integration time steps for intra-molecular and inter-molecular forces of 1 fs and 2 fs, respectively. Two identical 20 ns simulations were run starting from the same structure, but differing by the attribution of random initial atom velocities. Intermediate structures were saved every 250 ps. Dihedral angle analysis and generation of figures was carried out with scripts run within Pymol (https://www.pymol.org/).

## AFM imaging

Purified protein was diluted 10 to 50-fold with 10 mM NaPi, 300 mM NaCl to pH 6. Typically, 2 µL of the solution was then dispensed onto freshly-cleaved mica and proteins were allowed to adsorb for longer than 10 min. Samples were imaged after dilution with 150 µl of the same buffer. Standard image analysis and treatments were initially performed using NanoScope Analysis software (Bruker). When necessary, AFM images were processed with 0 to $3^{th}$ order plane fitting and 0 to $3^{rd}$ order flattening to reduce XY tilt. For further details on experimental approach and instrumentation, please refer to [23].

## Protein crystallization

A CcmK3/CcmK4 purified hetero-hexamer at 2 mg/mL was 5-fold diluted in buffer D (10 mM Tris, 20 mM NaCl, pH 7.5) and injected into a MiniQ PE 4.6/50 column at 0.5 mL/min rate on an AKTA purifier FPLC. Elution was performed with a 20 mL linear gradient from 0 to 80% buffer E, at 1 mL/min. Buffer E consisted of a mixture of 1,2-ethylenediamine (9.1 mM), 1-methylpiperazine (6.4 mM), 1,4-dimethylpiperazine (13.7 mM), Bis-Tris (5.8 mM), hydroxylamine (7.7 mM) and NaCl (300 mM) adjusted to pH 5.1 with HCl [36]. Immediately after elution, fractions (0.5 mL) were supplemented with 10 µL of 1 M HEPES pH 7.5. Only the two fractions following the most intense peak were pooled together. After overnight dialysis at 4˚C, overnight, against 500 volumes of 10 mM Tris, 150 mM NaCl, at pH 8, the sample was

concentrated to 2.9 mg/mL using Vivaspin 0.5 mL membrane concentrators (10 kDa MWCO).

Screening of crystallization conditions was performed using Mosquito drop dispensing automate (TTPLabtech) and crystallization screens (Qiagen). Drops were prepared by mixing 150 nL of protein solution with an equivalent volume of screening solution, at 12 ˚C. Crystals formed in drops prepared from reservoirs containing 22% (w/v) PEG 4000, 0.2 M ammonium sulfate and 0.1 M sodium acetate at pH 4.6. Crystals were briefly immersed in the reservoir solution supplemented with 20% ethylen-glycol before being cooled at 100 K in a cooled gaseous nitrogen flux. Diffraction data were collected on beamline ID30A3 at ESRF (European Synchrotron Radiation Facility, Grenoble, France) and processed using AUTOPROC (Global-Phasing, Cambridge, UK) and XDS.

The structure was solved using the molecular replacement method, as implemented in PHASER [37]. As the crystal potentially contained both CcmK3 and CcmK4, a model was built from the structure of CcmK4 (PDB entry 2A18), truncating all non-common side chains to alanine. The structure was refined with REFMAC [38] and COOT [39] from the CCP4 suite of programs [40], and deposited in the RCSB databank with PDB code 6SCR.

### Protein thermal denaturation studies

Differential Scanning Fluorimetry was used to characterize the thermal stability of selected homo- and hetero-heterohexamers. Mixtures of 20-µl of the sample (8–10 µM final, considering hexamers) and SYPRO Orange (× 100; Invitrogen) in Sol C were subjected to a temperature gradient from 20 to 100 ˚C with increments of 0.3 ˚C every 10 sec. Measurements were performed in triplicate in 96-well plates (Bio-Rad) with a real-time PCR CFX96 System (Bio-Rad). Melting temperatures ($T_m$) were extracted after adjustment of the full set of data from 2–3 experiments to a sigmoidal function using PRISM software, after normalization of fluorescence intensities.

Temperature-induced aggregation was evaluated using Dynamic Light Scattering. Assays were conducted on 10 µL of the protein sample in Sol C, applying a temperature gradient from 20 to 100 ˚C. Experiments were performed in a Zetasizer APS instrument (Malvern™, Panalytical Ltd., Malvern, UK). Scattered intensities were measured in function of the temperature (1 point/s, 2˚C/minute), and normalized. $T_{aggr}$ values were extracted after adjustment of the full set of data from 2–4 experiments to a sigmoidal function using PRISM software.

### Monomer exchange experiments

Untagged *Syn6803* K1 and K4 proteins were prepared by overnight treatment of TALON-purified His$_4$-tagged proteins following a described protocol [23]. After reaction, the solutions were flushed through TALON resins, which permitted to eliminate the TEV protease and possible unreacted protein traces. The flowthrough was collected, buffer exchanged against solution C, and concentrated to 1–2 mg/mL using 10 kDa MWCO concentrator units.

Subunit exchange studies were carried out in two ways. First, purified His$_4$-K4/K3-FLAG (7 µM final conc, 25 µL total volume) was conditioned in solutions with 50 mM NaPi (at pH 9 or pH 7) or NaAcetate (pH 5), 100 mM NaCl and 100 mM ammonium sulfate. In some cases, the mixture included 14% PEG 3350. In second type of experiment, the doubly-tagged His$_4$/ FLAG- homo- or hetero-hexamer (2 µM final conc, 25 µL total volume) in 50 mM NaPi (pH 8.0), 100 mM NaCl, and 0.15 mg/mL (final) of BSA (added to limit protein losses) was incubated in the absence or presence of untagged K1 or K4 (6 µM). Incubations were performed at room temperature, overnight. After addition of Tris buffer (150 mM final, pH 8), the TALON-resin was added (25 µL of 50% slurry, preconditioned in Sol A). The mixture was maintained

at 4°C, 5 min, with periodical shaking. After spinning down and removing supernatant to approximately dryness, 20 μL of a mixture containing LD (x1.5 final), EDTA (10 mM) and imidazole (300 mM) was added to the resin, and the mixture was heat denatured, before proceeding to WB analysis, as described.

## Results

### Selection of engineered *Syn6803* CcmK constructs

Prior to studying the compatibility between the different *Syn6803* CcmK components, we established the expression and solubility profiles of untagged proteins in *E. coli*. Collected data proved good expression of all 4 paralogs. Abundant soluble material was present after lysis in all cases, exception made of CcmK3 (abbreviated K3, S1 Fig).

The impact of tagging was also examined. Short peptides were selected and placed at either the N- or C-terminus. Overall, C-ter tagging was better tolerated. Thus, CcmK1 (K1) and CcmK2 (K2) were expressed and remained soluble regardless of the peptide tag identity at C-terminus (S2 Fig). Multiple bands observed for CcmK4 (K4) pointed to proteolysis phenomena. At the N-terminus, expression was restricted to $His_4$ and FLAG constructs, and only K2 and K4 remained soluble. Once again, no band corresponding to soluble K3 could be detected, irrespective of peptide type and tagging side.

### Hetero-hexamer formation between *Syn6803* CcmK paralogs

To investigate the potential association of paralogs within hexamers, all possible couples of genes coding for CcmK proteins were engineered in a pET26b-based plasmid that included two cassettes for independent expression (each one presenting T7 promoter/lac operator, RBS and T7 terminator) (S3A Fig). Combinations of $His_4$-tagged CcmK and a second FLAG-tagged CcmK were assayed in *E. coli*, following standard IPTG induction protocols. Bearing in mind results from previous section, plasmids coding for C-ter tagged K1, K2 and K4 were favored. Combinations with N-ter $His_4$-tagged K4 were also treated to anticipate the possibility that purified yields with the C-ter tagged K4 were low due to mentioned proteolytic instability of this construct. Finally, N- or C-ter FLAG-tagged and untagged K3 were also studied, in view of the demonstrated insolubility of the K3 paralog.

Cell contents, proteins remaining soluble after lysis and centrifugation, and purified fractions recovered on TALON resins were analyzed on Coomassie-stained SDS-PAGE gels (S3B Fig). Bands indicative of the occurrence of species with slightly different apparent size permitted to directly infer co-purification of K1 and K2, irrespective of which of the two was carrying the $His_4$ or FLAG tag. Double bands were also noticed when K4 labeled at the C-ter with $His_4$ and FLAG were expressed together. Absence of proteins was noticed in purified fractions for combinations with K3. This was the case for all 5 combinations with K3-$His_4$ (S3 Fig, 3rd lane). More surprising, purified proteins were also absent for combinations between K1-$His_4$ and FLAG-tagged or untagged K3 (white arrows). Despite less affected, K2-$His_4$ and K4-$His_4$ levels also diminished when co-expressed with K3, as compared to material purified in combination with other CcmK paralogs. The most intense bands in combinations with K3 were those of $His_4$-K4 (S3 Fig, 5th lane).

The presence of FLAG-tagged partners in purified samples was next verified by western blot (WB, Fig 1). Cases combining $His_4$- and FLAG-tagged versions of the same paralog served as positive controls (underlined in the table of S3A Fig), providing an indication of signal level attained for homo-hexamers. Such bands were clearly detected for K1-$His_4$/K1-FLAG, K2-$His_4$/K2-FLAG, K4-$His_4$/K4-FLAG and $His_4$-K4 /K4-FLAG. Signals revelatory of hetero-hexamer formation were intense for combinations between K1 and K2 (1st and 2nd lanes and

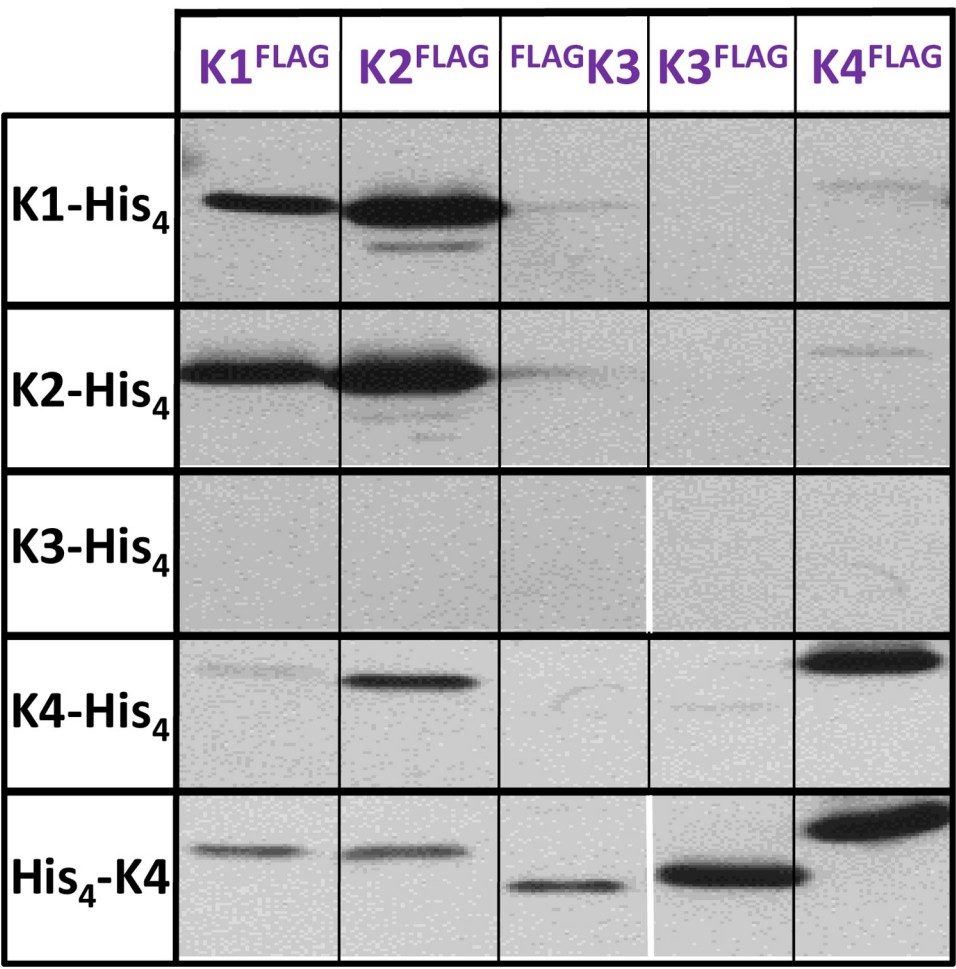

**Fig 1. Hetero-hexamer formation with combined *Syn6803* CcmK paralogs.** Western blot analysis of TALON-purified fractions. After running SDS-PAGE gels, and transferring proteins onto PVDC membranes, detection was performed with a mouse antiFLAG primary antibody followed by a secondary IgG antimouse-alkaline phosphatase fusion. The figure was prepared from two independent western blots that were performed in parallel. White vertical lines are to highlight the resulting image discontinuities.

columns in Fig 1), irrespective of which of the two carried the His$_4$ or FLAG peptide. Signals were comparable to those obtained for positive controls (K1 or K2 homo-hexamers). The most remarkable observation was, however, the detection of an intense band when His$_4$-K4 was combined with K3-FLAG. The intensity was, however, weaker than that observed for the K4 homo-hexamers. Some degree of cross-association was noticed between His$_4$-K4 and the FLAG-K3 construct, but also with FLAG-tagged K1 and K2, the latter being confirmed by a moderate intensity band with K4-His$_4$. Not surprisingly, signals were absent in combinations between His$_4$- and N- or C-ter FLAG-tagged K3 constructs, which also failed to reveal any purified material in Coomassie-stained gels (S3 Fig).

## Hetero-hexamer formation confirmed by native MS

BMC-H proteins are prone to assemble into large patches, observed *in vitro* but also during recombinant expression inside cells. Therefore, data presented in the previous section might

be explained as being the consequence of the purification of mixed assemblies combining different homo-hexamers. To clarify this possibility, three type of experiments were carried out. First, purified fractions that according to WB data contained hetero-hexamers (i.e. combinations between K1 and K2, or between His$_4$-K4 and K3) were analyzed by size-exclusion chromatography (SEC). The His$_4$-K4/K3-FLAG sample eluted at volumes characteristic of homo-hexamers, whereas K1-His$_4$/K2-FLAG showed intermediate behavior between hexamers and dodecamers. This elution behavior had been observed before for *Syn6803* K2 homo-hexamers, and is assumed to arise from formation of dodecamers consisting of stacked hexamers [15, 23, 41]. Second, cells expressing separately each homo-hexamer (His$_4$- or FLAG-tagged) were mixed up before lysis and purification. Fractions recovered from TALON resins did not reveal bands of FLAG-tagged proteins in WB. These experiments ruled out the copurification of assembled homo-hexamers, also the occurrence of monomer exchange between homo-hexamers during manipulations.

Finally, samples were inspected by native electrospray ionization mass spectrometry (native ESI-MS), an approach that is well suited for the characterization of molecular associations present in solution. This technique was exploited in a previous study to characterize His$_4$-tagged CcmK homo-hexamers [23]. After improving the molecular homogeneity of TALON-purified fractions by TEV treatment, hetero-hexamers could be detected in the 3500–5000 m/z range (Fig 2). For K1-His$_4$/K2-FLAG, the presence of the K2 paralog was directly evidenced by its two typical charge state distributions (CSD) at m/z 4000–5000 and 5500–6200 m/z. The second CSD, however, corresponded to species of lower MW than those detected before for the K2 homo-hexamer [23]. A zoomed view of the hexamer distribution indicated that every charge state was in fact split into several signals, matching to hexamers of different stoichiometries (shown in the inset of Fig 2A). After averaging over all charges, the calculated MW matched to K1/K2 stoichiometries ranging from 1:5 to 4:2 (S1 Table). Deviations from values calculated for combinations of monomers were small, below 50 Da. Such monomers were seen in the 1000–2000 m/z range, with masses indicative of loss of the first methionine, something usual for C-ter tagged proteins. Definitely proving the occurrence of hetero-hexamers, the two CcmK monomers were produced when given hexamer species were subjected to collision induced dissociation (CID) (Fig 2A, bottom panel). In addition, no evidence from K2-derived species could be obtained when similar experiments were performed on material purified from pools of cells expressing K1-His$_4$ and K2-FLAG separately (S1 Table).

Analogous conclusions were drawn from data collected on complexes formed between His$_4$-K4 and K3-FLAG (Fig 2B). The most important evidence was the detection of the two monomers dissociating from selected hexamer species submitted to CID. Monomer masses were in excellent agreement with the expected value for TEV-untagged His$_4$-K4 and K3-FLAG (after loss of first methionine with the latter). The attribution of hexamer peaks in the 3500–5000 m/z region to different stoichiometries was less straightforward than for K1/K2 complexes, a likely consequence of the small MW difference between the two monomers (113 Da, or 244 Da if K3 had lost the first Met). The most intense signal best matched a K3/K4 hetero-hexamer with 1:5 stoichiometry. K4 homo-hexamers were also detected, something that contrasts with data for K1/K2 that only revealed hetero-hexamers. It is also worth mentioning that faint signals were detected for a 71.1 kDa species. This MW closely matches a hypothetical K3 hexamer, which should however not be retained by the TALON resin. Alternatively, they could derive from homo- or hetero-hexamers composed of partially degraded CcmK subunits. Two other minor species, of approx. 37.7 and 66.8 kDa, could not be attributed to any n-mer combination.

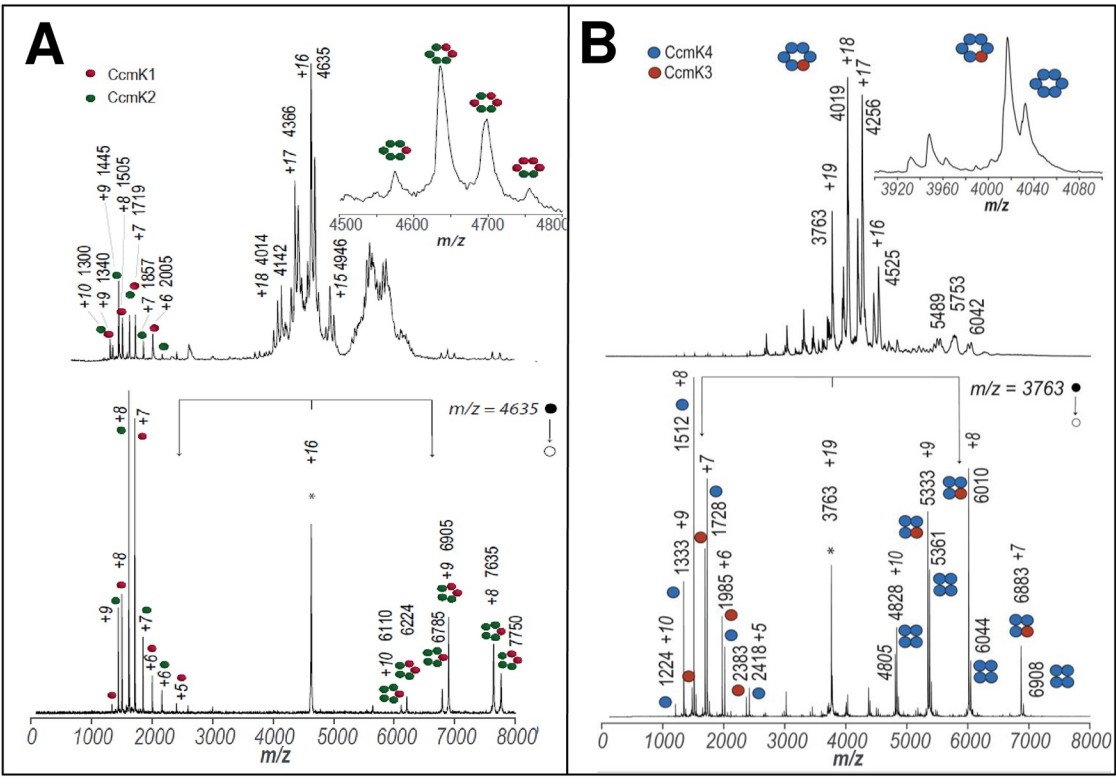

**Fig 2. Hetero-hexamer characterization by native mass spectrometry.** *A*, Spectra recorded on fractions purified from cells co-expressing CcmK1-His$_4$ together with CcmK2-FLAG; *B*, similarly, but from cells co-expressing His$_4$-CcmK4 and CcmK3-FLAG. Positive-ion mode native ESI-MS spectra are presented on top panels. These spectra are dominated by signals from multiply charged ions of hexamers. Potential higher order assemblies were observed in combinations with CcmK2. Bottom panels present an example of MS-MS collisional activation data collected on hexamer precursor ions selected from top spectra (indicated with an asterisk). An asymmetric charge partitioning is noticed, hexamers dissociation resulting in two type of monomers and either pentameric species with CcmK1/K2 (left) or tetrameric species with CcmK3/K4 (right). Monomer masses permitted to attribute picks to either CcmK1 (red spheres), CcmK2 (green), CcmK3 (orange) or CcmK4 (violet). Species m/z values and charges are indicated for major peaks. Cartoons schematically illustrate the stoichiometries of detected protein complexes and subcomplexes. Molecular weights of neutral species estimated from data from different charge states are compiled in S1 Table. Please notice that tags were removed prior to spraying by treatments with TEV protease.

## Simultaneous expression of all CcmK paralogs

Co-expression of CcmK couples proved the structural compatibility between CcmK1 and CcmK2 or between CcmK3 and CcmK4 couples. This coincidently reflects the organization of each CcmK couple at separated chromosomal loci, and might therefore support an independent evolution of each CcmK pair of sequences. Our data, however, did not rule out the possibility of attaining more complex associations in situations of concomitant expression of all four paralogs. Transcriptomic data indicate that all four CcmK paralogs might be expressed simultaneously in *Syn6803*, depending on environmental conditions [42]. Besides, weak WB signals noticed for combinations between K4 and FLAG-tagged K1 and K2 (Fig 1) suggested that other combinations might lead to more complex associations.

A similar strategy as presented above was adopted to investigate this point, the main difference being that engineered plasmids included four cassettes permitting independent expression of proteins tagged with different short peptides: His$_4$, StrepTag, FLAG and HA at cassettes 1, 2, 3 and 4, respectively. We limited our screening to eight combinations (schematized in S4A Fig). These included His$_4$ C-ter tagged K1, K3 and K4, as well as N-ter tagged

His$_4$-K4 for reasons mentioned above (all in cassette 1). Depending on the identity of the paralog at first cassette, N- or C-ter FLAG tagged K3 or K4 were placed at cassette 3, and either K1-HA or K4-HA at cassette 4. All cases included K2-StrepTag at cassette 2. Additionally, a positive control was assayed that consisted of K2-His$_4$ combined with K2 labeled with all three other peptides at cassettes 2 to 4, and which should inform on signal thresholds attained with identical configurations leading to homo-hexamers.

The co-expression of the different combinations was screened in *BL21(DE3)* cells after classical induction with IPTG or under auto-induction conditions. Six of the eight screened plasmids resulted in sufficient purified protein as to be visualized in Coomassie-stained SDS-PAGE gels. Most intense bands were those obtained with the two plasmids combining His$_4$-K4/K2-Strep/FLAG-K3/K1-HA and His$_4$-K4/K2-Strep/K3-FLAG/K1-HA (S4B Fig, cases 7–8), which revealed bands irrespective of the choice of induction system. Moderate intensity bands were produced with the K3-His$_4$/K2-Strep/FLAG-K4/K1-HA case (case 3). Bands were faint but more clearly detected under auto-induction conditions with K1-His$_4$/K2-Strep/ K3-FLAG/K4-HA (case 2), K3-His$_4$/K2-Strep/K4-FLAG/K1-HA (case 4) and K4-His$_4$/ K2-Strep/K3-FLAG/K1-HA (case 6). Most unexpected were bands recovered in combinations with K3-His$_4$, since this construct is normally insoluble and reluctant to purification, as shown in previous experiments (S1 and S2 Figs). Moreover, the major band for K3-His$_4$/K2-Strep/ FLAG-K4/K1-HA corresponded to the FLAG-K4 partner, indicating a low incorporation of the K3-His$_4$ within purified hexamers, in good agreement with the stoichiometries identified by native ESI-MS (see above). None or very faint bands were revealed for the two combinations implying the K1-His$_4$ partner (S4B1 Fig), which is otherwise well soluble and purified in good amounts when expressed alone (S2 Fig). We hypothesized that this could be due to interferences caused by K3, as indicate results collected in studies of co-expression of K1/K3 couples (S2 Fig).

Inspection by WB identified K3/K4 and K1/K2 as being the dominant associations. Thus, FLAG signals were intense for combinations implying His$_4$-K4 and K3-FLAG and for K3-His$_4$ with FLAG-K4 (Fig 3, see also S4C Fig). Less intense associations were revealed for K3/K4 combinations with tags on the same side. The only signal corresponding to StrepTag or HA in purified fractions was revealed for the combination K1-His$_4$/K2-Strep/K3-FLAG/K4-HA. The low signal level, which again supported a K1/K2 interaction, can be explained by the low amount of purified protein. Validating our approach, intense signals from StrepTag, FLAG and HA peptides were detected for the purified homo-hexamer control deriving from simultaneous expression of K2-His$_4$, K2-Strep, K2-FLAG and K2-HA (Fig 3B, last column).

## Sequence-structural considerations for the formation of CcmK3 homo-hexamers or CcmK3/K4 hetero-hexamers

Despite well expressed, the low solubility of *Syn6803* K3 precluded the investigation of its solution behavior or attempts to elucidate its crystal structure. To understand whether this behavior could be linked to an incorrect folding of the K3 monomer or to the presence of residues incompatible with oligomerization, the amino acid sequence was first aligned with those of other *Syn6803* paralogs or with CcmK from *Syn7942* and *Hal7418* species recently studied by Kerfeld and coworkers [34] (Fig 4A). Identity scores were considerably lower when considering members of the K3 family (56% in average) than within all other CcmK (mean 90% and 68% for K1/K2 and K4 families, respectively) (S2 Table). The conservation raised significantly when comparing positions implicated in inter-monomer contacts in CcmK1/K2 and K4 (96 and 81% average identities, respectively), as noticed before [43]. However, the increase almost vanished when residues at the presumed K3 monomer interfaces were compared (58%).

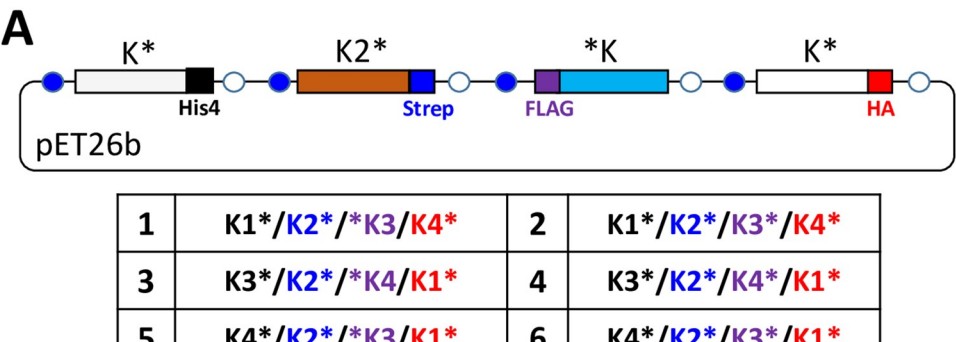

**Fig 3. CcmK hetero-association occurrence upon co-expression of all *Syn6803* CcmK paralogs.** *A*, Schematic representation of screened vectors. Four ORFs coding for CcmK proteins tagged at either N- or C-terminus with His4 (black), Strep (blue), FLAG (violet) or HA (red) peptides were engineered between T7 promoter and terminator sequences (blue and empty circles, respectively). Studied combinations are listed in the table below. Whether asterisk is written on before or after the paralog name denotes N- or C-tagging emplacement, respectively. *B*, Western blot analysis of TALON-purified fractions, after 10-fold concentration using 10 kDa MWCO filter devices. Studied cases were those developing bands in purified fractions (see S4C Fig) and are indicated on the top following the numeration of the table. The last lane corresponds to a CcmK2 homo-hexamer purified from a strain transformed with a K2-His4/K2-Strep/K2-FLAG/K2-HA co-expression vector, which confirmed the presence of all tags. Detection on PVDC membranes was effected with either streptactin-AP conjugate (first horizontal lane), or with either mouse antiFLAG (second) or mouse antiHA (third) followed by final revelation with a secondary IgG antimouse-AP fusion. The bottom lane was prepared after continuing the incubation of the membrane shown in the second lane with a mixture of all three detection systems, followed by antimouse-AP and revelation. Please notice that several raw data images needed to be spliced and rearranged to prepare this figure.

The most remarkable difference, when comparing *Syn6803* K3 to other CcmK paralogs, is the presence of the two ionic bulky amino acids Glu38 and Arg39 (Fig 4A) [34]. These residues, also present in *Hal7418* K3 but changing to Glu38 and Ser39 in *Syn7942*, would likely clog the central hole of a potential CcmK3 hexamer. Also in common with the *Hal7418* K3, potential secondary structure-disrupting prolines are found in *Syn6803* K3 (Pro49 and Pro83). Other notable substitutions, specific of *Syn6803* K3, are the replacement of a few small/

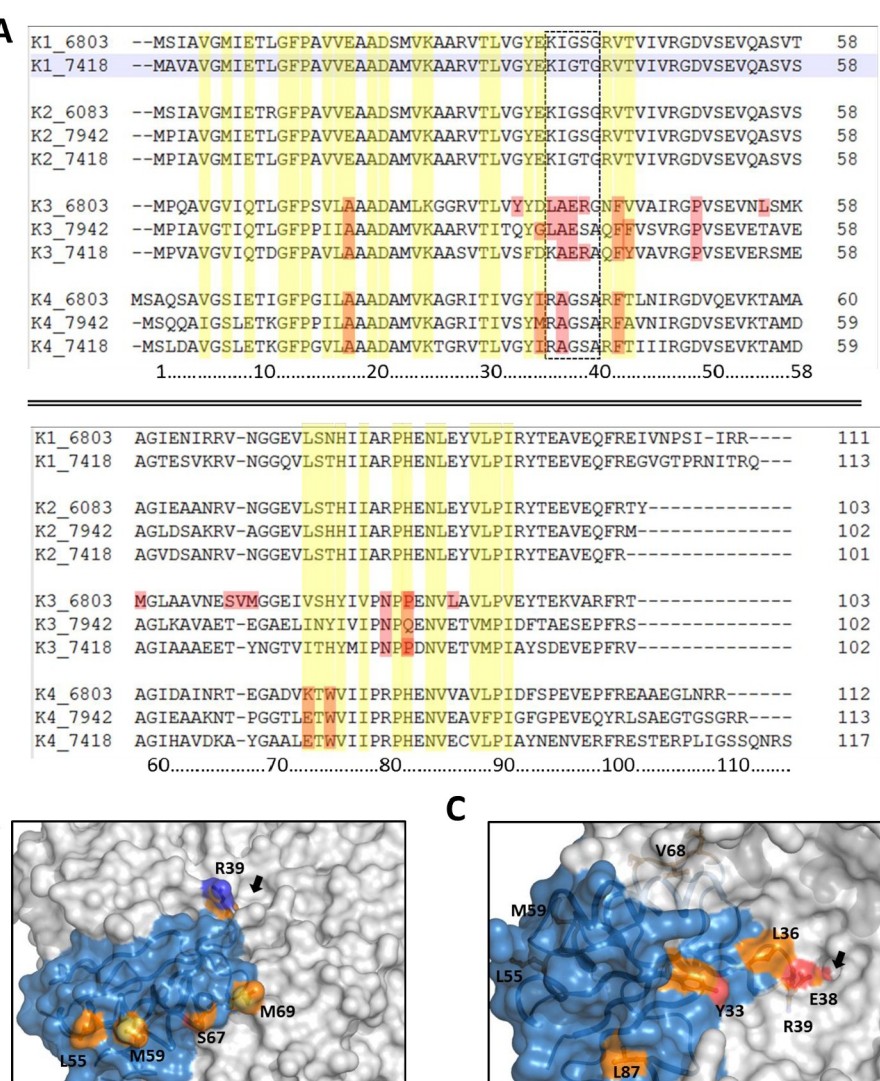

**Fig 4. Sequence and structural considerations on CcmK3 paralogs.** *A*, Sequence alignment of sequences of CcmK from *Syn6803, Syn7942 and Hal7418*. Residues directly participating in inter-monomer contacts in 3D structures of homo-hexamers are shaded in yellow. In red are highlighted notable substitutions present in Syn6803 CcmK3, which differ considerably in other paralogs and can be shared or not by CcmK3 in other species. The rectangle contoured by discontinuous traces indicate residues that line central CcmK hexamer pores. *B, C* Presumed localization of some notable residue substitutions in the CcmK3 *Syn6803* structure furnished by SWISS-MODEL (CcmK3_1 in the main text), when compared to other CcmK. The surface and ribbon representation are shown with one chain colored light blue, neighbor subunits in grey. Panel B presents the water-exposed orientation of several hydrophobic side-chains residues localized on the predicted α-helix-2 of CcmK3 (balls and sticks representation, with carbon atoms in orange, nitrogens in blue, oxygens red and sulfur atoms in yellow), as well as Glu38-Arg39 residues lining the central hole. The helix runs from the hexamer edge to the center, and is followed by the stretch comprising S67 to M69, where a residue insertion specific of Syn6803 CcmK3 occurs. The emplacement of the hexamer clogged central hole is indicated by the arrow. The model is viewed from the convex face. Panel C shows the emplacement of other diverging residues, like the Tyr33, the Leu87 or the pore-clogging Glu38-Arg39 residues. The hexamer model is viewed from the concave face and the emplacement of the clogged central pore is indicated by the arrow. Please notice that the side-chains of Arg39, Leu55, Met59 and Val68 lie on the opposite face.

hydrophilic residues by the bulkier/hydrophobic Tyr33, Leu36, Leu55, Met59, and Leu87, and a residue insertion in the stretch composed by Ser67 to Met69.

To further visualize these differences, homology models were built using SWISS-MODEL (hereafter CcmK3_1 model) and PHYRE2 (CcmK3_2) algorithms. The resulting monomer models were similar, with average root-mean-squared-deviation (RMSD) of about 0.6 Å calculated for the position of 246 backbone atoms (excluding the C-ter 94–103 residues). Deviations increased slightly for recomposed hexamers (RMSD of 1.0 Å estimated for 1508 backbone atoms). Structural differences were most remarkable for three stretches (S5A Fig): residues 37 to 42 around the hypothetical central hexamer pore; the region comprised between Glu66 and Gly71 that connects helix-2 and strand-4, where the mentioned single residue insertion occurs; and the region delineated by residues Leu87 to Pro91. Besides, different conformers were proposed by the two algorithms for the side-chain of Tyr77. Similarly, the side-chain of Tyr33, which corresponds to Gly or Ser in other paralogs, was also rotated towards the protein surface in CcmK3_1, but pointing towards the monomer core in CcmK3_2. Several hydrophobic residues were also predicted to lie in the water-exposed side of an α-helix (e.g. Leu55 and Met59, which are Ala or hydrophilic residues in other sequences) on the hexamer convex surface (Fig 4B), something that would increase the aggregation trend of *Syn6803* CcmK3.

Without evident reason to favor one model over the other, the two were exploited to construct hetero-hexamers combining 5 K4 units and a single K3 monomer, and their behavior was investigated by molecular dynamics simulations, in comparison to K3 homo-hexamers (S5B Fig). The two K3/K4 hetero-hexamer models built from CcmK3_1 or CcmK3_2 displayed similar structural robustness, rearranging slightly over the first nanoseconds but remaining basically unchanged for the rest of the simulation. Remarkably, a closer inspection indicated that the conformation of Tyr33 and Try77 side-chains of CcmK3_2 model rearranged over the course of simulations to adopt a similar disposition as in the CcmK3_1 monomer. In agreement with these data, the CcmK3_1 homo-hexamer model seemed structurally stable, contrasting with the CcmK3_2-based homo-hexamer that failed to reach convergence. Results were virtually identical in two independent 20 nanosecond MD simulations runs that only differed by the attribution (random seed) of initial atom velocities.

A visualization of modeled structures suggested some interactions that might contribute to contacts between CcmK3_1 and CcmK4. One example could be an ionic interaction between the side-chains of CcmK4 Arg38 and CcmK3 Asp35 (S6A4 Fig). Such interaction would reproduce what is found in CcmK1 and CcmK2 hexamers (S6A1 Fig). On the contrary, it is abrogated in CcmK4 (S6A2 Fig) or in a hypothetical CcmK3 homo-hexamer (S6A3 Fig) as a consequence of the replacement of one of the two ionic residues by hydrophobic amino acids. Coincidently, the couple found at the respective positions of the second inter-monomer face of CcmK3 might be advantageous too: CcmK3 Leu36—CcmK4 Ile37 (S6A8 Fig). In CcmK1, an ionic pair is found at the corresponding positions (S6A5 Fig), whereas combinations of hydrophobic plus ionic residues occur in CcmK4 or in modeled CcmK3 hexamers. Similarly, the combined presence of His76 of CcmK3 and Glu70 of CcmK4 might be beneficial (S6A12 Fig).

The electrostatic properties calculated for the CcmK3_1-based homo-hexamer are well different from those of other paralogs (S6B Fig). The isoelectric point (pI) estimated by PropKa for the CcmK3_1 structure is 7.0, which compares to pI 6.0 and 5.2 for CcmK1 and CcmK4, respectively. The most striking difference was noticed on the electrostatic surface potential from the concave face side, which seemed inversed with regard to those from other paralogs at neutral pH, as pointed out by Sommer *et al.* [43]. The differences were attenuated when a single CcmK3 monomer was modeled within the CcmK4 assembly (pI 5.3 estimated for the hetero-hexamer). Yet, mostly caused by the presence of K3 Glu38 and Arg39, the physical

properties of the hetero-hexamer pore would be substantially modified, and the hexagonal symmetry broken.

Overall, the consideration of 3D homology models and results of MD simulations suggest that the stability of CcmK3 might be higher when embedded with CcmK4 than in homo-hexamers. Furthermore, the augmentation of the hydrophobicity of the *Syn6803* CcmK3 monomer surface, which would cumulate in a homo-hexamer, might be alleviated when combined to CcmK4 in hetero-hexamers.

## Structural investigations of *Syn6803* CcmK3/K4 hetero-hexamers

Two experimental approaches were followed to characterize structurally the K3/K4 hetero-hexamer behavior. First, its assembly behavior was monitored by atomic force microscopy (AFM) and compared to the K4 homo-hexamer (Fig 5A, left). Despite formation of 2D patches noticed in some experiments for K3/K4, the assemblies were smaller and less regular than those obtained with $His_4$-tagged K4 homo-hexamers (Fig 5A, right), which basically reproduced previous published results [23]. Assembly plane with K3/K4 samples positioned about 2.8 nm above mica surfaces, though this value is likely an under-estimate induced by high surface coverages. The degree of organization on mica of CcmK1/K2 hetero-hexamers was even lower. Images revealed only the presence of individual hexamers, of clusters of variable size and of some linear arrangements (Fig 5B). We hypothesize that the decreased assembly potential of the last sample could be in part due to its high molecular heterogeneity, as shown by the variable stoichiometries determined by native ESI-MS.

The crystallization of K3/K4 hetero-hexamers was next attempted. We opted to increase K3/K4 abundance by means of pIsep-FPLC, a chromatofocusing-like purification approach (Fig 5C). Our intention was to exploit the expected strong interaction with the cationic resin of the anionic DYKDDDDK FLAG peptide in fusion to K3. In that manner, the presumed $His_4$-tagged K4 homo-hexamers could be removed, as indicated by the WB analysis of purified fractions that proved that most of FLAG signal was eluting with the second (most abundant) chromatographic peak. The pooled fractions collected for this peak were used for crystallographic assays.

A condition resulted in a hexagonal crystal form diffracting X-rays to 1.80 Å resolution and displaying two molecules in the asymmetric unit. Unexpectedly, although both CcmK3 and CcmK4 were present in the crystallization solution, the corresponding refined structure unambiguously showed the sole presence of K4 subunits in the asymmetric unit, which organize as canonical homo-hexamers as the result of space group symmetry (structure deposited with PDB code 6SCR, statistics presented in the S3 Table). The monomer structures were basically identical to previously solved structures, with only 0.25 and 0.23 Å RMSD for 408 backbone atoms, when compared to structures deposited in the RCSB databank with PDB codes 2A18 and 2A10, respectively. The only novel feature was the observation of a 2D arrangement with similarly-oriented K4 hexamers (Fig 5D), which therefore differs from the stripped organization described before for the same protein [13], and provides an additional proof of the high structural plasticity of BMC-H proteins.

## Evaluation of *Syn6803* CcmK hetero-hexamer stability

CcmK3 absence in crystals prepared from samples of K4/K3 hetero-hexamers pointed to monomer rearrangements occurring during crystallization. To shed light on this possibility, purified $His_4$-K4/K3-FLAG was first incubated overnight at variable pH, and under conditions resembling those that resulted in the formation of CcmK4 crystals, and FLAG signals remaining associated to the $His_4$-tagged component were quantified by WB after sedimentation of

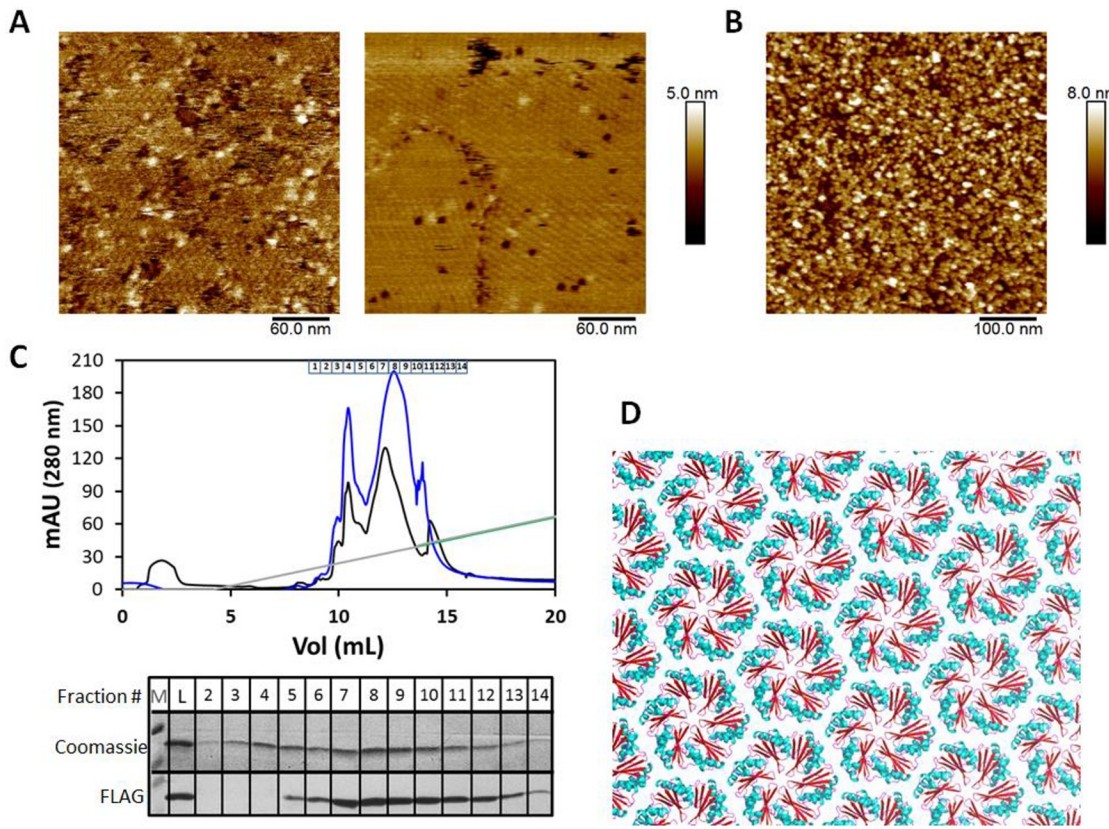

**Fig 5. Structural analysis of *Syn6803* CcmK hetero-hexamers.** AFM images recorded after adsorption on mica of 70 ng of CcmK3/K4 (panel *A*, *left)* or180 ng of CcmK1/K2 hetero-hexamers *(B)*, in sodium phosphate buffer at pH 6.0. For comparison, 2D arrangement obtained under similar conditions after deposition of 70 ng of CcmK4 homohexamer *(A, right)*. *C*, pIsep-FPLC purification chromatogram after injection of His$_4$-CcmK4/CcmK3-FLAG hetero-hexamer sample (two different runs are shown, in black and blue trace). Elution was performed with a linear gradient of a polybuffer solution at pH 5 (gray line). Approximate fraction emplacements are indicated above the chromatogram. Below is shown the 10 to 15 kDa region after migration of different fractions on a SDS-PAGE gel. Please notice that His$_4$-CcmK4 and CcmK3-FLAG bands could not be resolved. FLAG-signals detected by WB are shown below. Bands were delayed with regard to Coomassie-stained bands. Pooled fractions 8 and 9 were selected for crystallographic experiments. *D*, Crystallization of that pooled fraction permitted to solve structures that revealed the sole presence of CcmK4. Shown is the resulting CcmK4 crystal packing, revealing a molecular layer with hexamers arranged following a uniform orientation (shown with CcmK4 convex side facing up). Interhexamer spacing is 7,1 nm. A ribbon view is presented, with elements colored in accordance to secondary structure elements.

material retained bound to TALON beads. These experiments failed to reveal any signal drop (Fig 6A, top), independently of the incubation pH or of the presence of the PEG additive, which was included at lower concentrations than for crystallization assays in order to limit protein precipitation. In a similar experiment, we sought to enhance monomer exchange by incubating the K3/K4 hetero-hexamer (combining His$_4$ and FLAG monomers) with a 3-fold molar excess of untagged K4 or K1 homo-hexamers. Exchange of K3-FLAG monomer for an untagged subunit was expected to result in a diminution of FLAG readings. However, K4/K3 was as stable as the K4 homo-hexamer, signals remaining similar for incubations carried out in the absence or in the presence of untagged hexamers (Fig 6A, bottom).

Absence of monomer exchange confirmed the consensual vision of BMC-H components as being robust. Similar conclusions had been made in other studies that were conducted on homo-hexamer mixtures and monitored by crosslinking high-mass MALDI-MS or native ESI-MS [44]. Nevertheless, we decided to compare the thermal stability of hetero- *vs* homo-

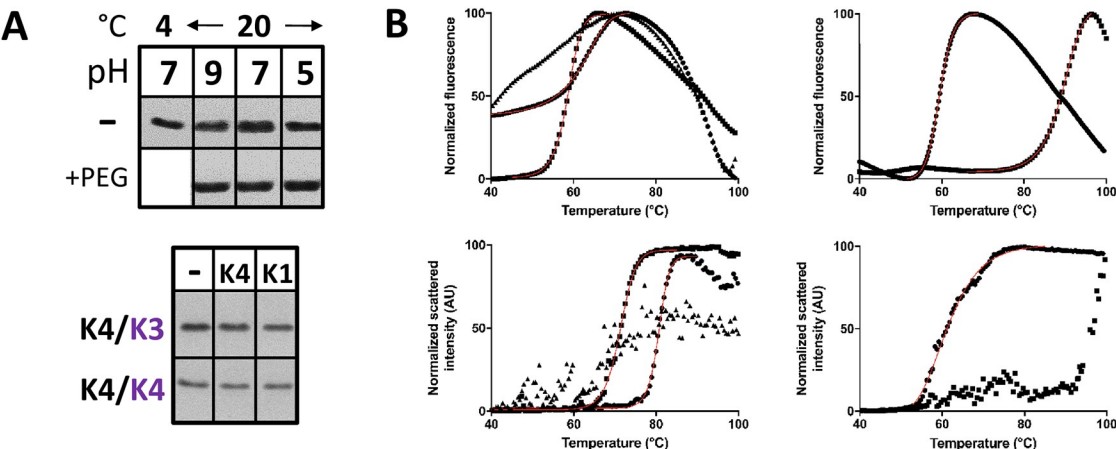

**Fig 6. Stability of *Syn6803* CcmK hetero-hexamers.** *A*, Subunit exchange investigated by western blots. *Top*, FLAG signals recovered bound to TALON beads after reacting with purified His$_4$-K4/K3-FLAG (7 μM monomer concentration) that had been pre-incubated overnight, at pH ranging between 5 and 9, in the absence (top lane) or presence of 14% PEG 3350 (bottom). The first lane corresponds to a sample at pH 7 that was maintained at 4˚C throughout the overnight incubation. The bottom panel presents FLAG signals detected for indicated homo- and hetero-hexamers combining His$_4$- and FLAG-tagged subunits (2 μM final monomer concentration, indicated by black or violet letters, respectively) that were incubated overnight at room temperature and pH 8 in the absence or in the presence of untagged K4 or K1 homo-hexamers (6 μM monomer). *B*, Thermal denaturation monitored by DSF (*top*) and DLS (*bottom*). Left panels present data collected for CcmK1 (triangles) and CcmK2 (circles) homo-hexamers, as well as for CcmK1/K2 hetero-hexamers (squares). Right panels show data for CcmK4 (squares) and CcmK3/K4 (circles). In DSF experiments, the proteins were incubated with the Sypro dye and the fluorescence of the probe was monitored as temperature was increased. For DLS assays, changes of the intensity of scattered light of hexamer solutions with temperature were recorded. DSF and DLS data were normalized before analysis. The mean of multiple measurement (see S4 Table) is shown with black symbols as well as the adjusted sigmoids (red lines). For each trace, only one every two recorded data points are displayed to facilitate figure interpretation.

hexamers by Differential Scanning Fluorimetry (DSF). This technique often permits the monitoring of protein unfolding processes that expose hydrophobic patches, causing an increase of the fluorescence of a probe. In that manner, *Syn6803* K1/K2 and K3/K4 hetero-hexamers exhibited denaturation profiles with strikingly similar midpoint melting temperatures ($T_m$) of 58.7 and 59.6 ˚C, respectively (S4 Table, Fig 6B). In comparison, $T_m$ of 89.2 and 62.3 ˚C were measured for K4 and K2, respectively, whereas fluorescence readings with the K1 homo-hexamer changed slowly and weakly with temperature, which prevented the determination of a $T_m$ value.

Thermal stability was further assessed by Dynamic Light Scattering (DLS), which is very sensitive to aggregation phenomena that often accompanies protein unfolding. Yet, care must be taken in interpreting results, especially considering that CcmK are subject to auto-assembly. Overall, measurements of the light scattered intensities upon augmenting the temperature produced similar trends as DSF and confirmed the lower stability of hetero-hexamers (Fig 6B). After fitting DLS data to sigmoid functions, $T_{aggr}$ of 71.3 and 61.3 ˚C were calculated for K1/K2 and K4/K3 hetero-hexamers, respectively (S4 Table). The value for the former was significantly higher than measured by DSF, suggesting that the fluorescent probe could have played a deleterious effect. On the contrary, values estimated by DSF and DSL for K4/K3 were similar. Most importantly, $T_{aggr}$ values were displaced towards higher values for homo-hexamers. Thus, a $T_{aggr}$ of 80.6 ˚C was calculated for K2. It is noteworthy that $T_{aggr}$ values could not be measured for K1 and K4, the light scattering intensity augmenting slowly and unstably in experiments with the first, or starting to occur when the T was above 90 ˚C with the second (Fig 6B).

## CcmK3 and CcmK4 paralogs from *Syn. elongatus* PCC 7942 also form hetero-hexamers

To investigate whether CcmK hetero-oligomerization might be common to other species, we applied the same experimental approach to the study of association between K3 and K4 paralogs of the model *Syn7942* β-cyanobacteria. Plasmids were prepared permitting the co-expression of both isoforms tagged at N- or C-terminus with $His_6$- and FLAG-peptides. Protein expression, solubility and purification in *E. coli* were assessed by the same means as for experiments with the *Syn6803* paralogs. Controls were set up to quantify signals obtained when the same paralog (CcmK4) was co-expressed tagged with both $His_6$ and FLAG tags. By the time of realization of these experiments, Kerfeld and colleagues published evidences on the formation of hetero-hexamers between *Syn7942* K3 and K4 [34]. We decided nevertheless to complete this portion of our study, especially because differences between the two experimental designs might permit the opportunity to obtain complementary information.

Bands corresponding to K3 co-purifying with $His_6$-tagged K4 were evident in Coomassie-stained gels (regardless of tag emplacement) (S7P Fig, bottom). K3 bands were nevertheless much fainter than those of K4-$His_6$, contrasting with similar intensities detected for cellular or soluble fractions (S7C/S7S Fig). The ratio of K3/K4 intensities was higher for $His_6$-K4 co-expressed with K3-FLAG, when induction was triggered with IPTG, but the effect can probably be attributed to a much lower expression of the $His_6$-K4 partner. An interesting observation was that, unlike *Syn6803* K3 that is basically insoluble, *Syn7942* K3 was still present in the soluble fraction when expressed alone, and could be even purified. Also noticeable, purified K4 bands were broad in the Coomassie gels, suggestive of potential degradation of the protein, somehow resembling observations on *Syn6803* K4.

The presence of K3/K4 hetero-hexamers was confirmed on WB. The intensity of FLAG bands in purified fractions was lower for samples containing K3-FLAG than in combinations resulting in K4 homo-hexamers (Fig 7). Taking into consideration that FLAG signals were

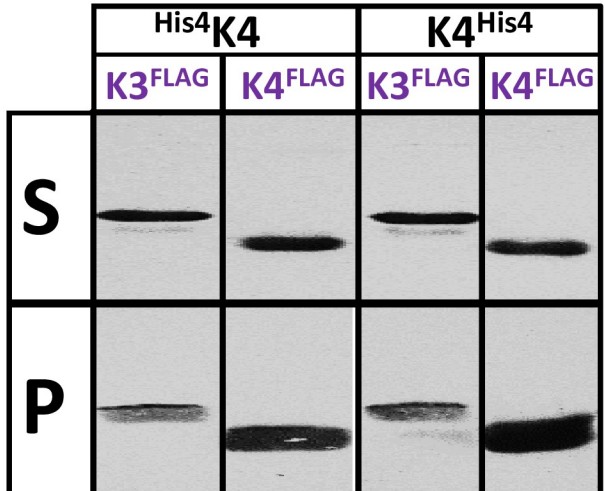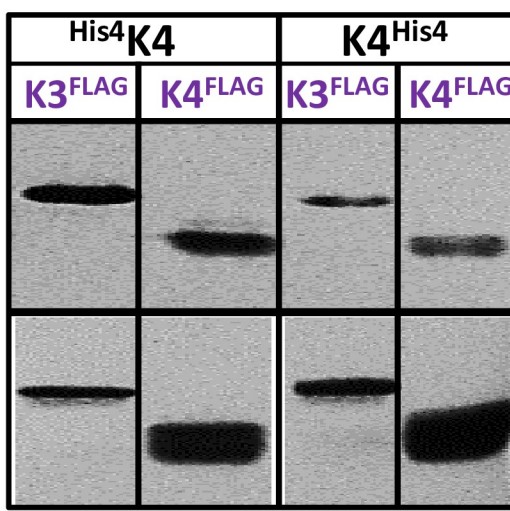

**Fig 7. Hetero-hexamer formation between *Syn7942* CcmK3 and CcmK4.** Western blot analysis of fractions remaining soluble after cellular lysis and centrifugation (top lane) and material purified on TALON resins (bottom). Left panel correspond to samples prepared from cellular cultures that were induced in IPTG (200 μM), whereas data on the right correspond to samples produced under auto-induction conditions. Detection on PVDC membranes was performed with a mouse antiFLAG primary antibody followed by a secondary IgG antimouse-alkaline phosphatase fusion. Two raw data images were spliced and rearranged to prepare this figure.

comparable for all soluble fractions, these data again suggest that the recruitment of K3-K4 might be suboptimal.

## Discussion

The structural characterization of BMC shells is important to understand function, also for engineering novel structures with properties tailored to new applications in synthetic biology. Among other properties, shell permeability, compartment robustness and adaptability to variable environmental conditions will depend on molecular properties of assembly "bricks". Until recently, shell components were presumed to be homo-oligomeric associations exclusively. This scenario, however, neglected the possibility that multiple protein paralogs present in a given organism could cross-associate to form heteromers.

Several lines of evidence advocated the formation of such hetero-associations. Bioinformatics surveys have proven the existence of multiple BMC-H (up to 15 genes), BMC-T (up to 5) and BMC-P (up to 7) in given organisms [3]. Very often, several paralog genes are found within a same operon, and are expected to result in simultaneous protein expression. Moreover, paralog homologies are high. Another indication might be the existence of regulatory mechanisms to ensure the assembly of a single type of BMC in the many organisms that are equipped with several BMC candidates [45]. An example is the repression in *Salmonella enterica* of transcription of the ethanolamine utilization (Eut) BMC by 1,2-propanediol (1,2-PD), the substrate (and inducer) of the propanediol utilization (Pdu) BMC [46]. In this study, phenotypic changes of growth on 1,2-PD, symptomatic of a disrupted BMC shell, were noticed when the microorganism was engineered to permit the unregulated expression of Eut proteins or of CcmO from cyanobacteria. We hypothesize that such regulatory mechanisms would serve to control the intrinsic structural promiscuity of BMC shell components, both at the monomer and hexamer levels.

This study demonstrate the structural compatibility between *Syn6803* K1/K2 and K3/K4 hetero-hexamers. Data were more consistent for the *Syn6803* K1/K2 combination, results being unaffected by the choice of tagging configuration (Fig 1). WB signal intensities were comparable to those measured for FLAG-labeled homo-hexamers, suggesting that K1/K2 might compete efficiently with homo-hexamer formation *in vivo*. Moreover, analysis of purified samples by native MS/MS, which confirmed the association of subunits within the same hexamer, permitted to detect almost all possible stoichiometries, supporting the good structural miscibility between the two paralogs.

Most remarkable was the purification of *Syn6803* K3 complexes in combination with K4, especially considering that K3 is recurrently found insoluble when expressed alone in *E. coli*. The formation of the *Syn6803* K3/K4 hetero-hexamer was first noticed for the combination between the K3-FLAG and $His_4$-K4. This observation was confirmed when the four CcmK were co-expressed together, which in addition demonstrated the viability of other combinations (i.e. K3-$His_4$/FLAG-K4). These experiments also highlighted the structural segregation of K1/K2 from K3/K4 paralogs, the two pairs of paralogs being coincidently encoded together but in separated operons of *Syn6803*. Thus, cross-associations between $His_4$-labeled K4 and K1 or K2 that insinuated as faint WB signals in studies with CcmK couples (Fig 1), were absent when all CcmK were expressed together, indicating that they might be irrelevant or only happen under given relative expression regimes.

Hetero-hexamers also formed upon co-expression of *Syn7942* K3 and K4, confirming and complementing conclusions reported recently on the association of K3/K4 from *Syn7942* or *Hal7418* [34]. In that study, a $His_6$/StrepTag tandem purification approach permitted to establish a 2:4 K3:K4 association stoichiometry, which we also detected for *Syn6803* paralogs

together with the most prominent 1:5 hexamer. In our study, we also sought to estimate the extent of competition between processes leading to hetero- *versus* homo-hexamers. Although the interpretation is still complicated by parameters such as relative expression/solubility levels of each paralog in *E. coli* or tag tolerance differences among CcmK paralogs, the overall trend indicated that K3/K4 formation is relatively inefficient. The conclusion is based on: i) weaker *Syn6803* K4/K3 WB signals, when compared to FLAG-tagged CcmK4 homo-hexamers; ii) Coomassie bands for K3-His$_4$ significantly less intense than those of copurifying FLAG-K4 (S4B Fig, case 3); iii) detection by native ESI-MS of CcmK4 homo-hexamers contaminating purified K3/K4 samples (also inferred by pISep-HPLC); iv) the most abundant stoichiometry of K3:K4 is 1:5. In our hands, the competing potential of *Syn7942* K3 seemed even lower, as evidenced the weaker WB signals for purified K3/K4 samples when compared to K4/K4 homo-hexamers (Fig 7), and the small K3:K4 ratio of Coomassie band intensities in K3/K4 purified samples, which contrasted with comparable or higher K3 intensities detected in soluble fractions (S7 Fig).

A close inspection of the position of *Syn6803* K3 and K4 interfacial residues pinpointed compensatory residue substitutions that might contribute cooperatively to the cross-association. However, we failed to get proofs in support of a co-evolution of pairs of K3/K4 residues using specialized algorithms (i.e. MISTIC or EVcomplex). Hetero-hexamer stability was supported by molecular dynamics trajectories of 3D hetero-hexamers built from combinations of a CcmK4 crystal structure and 3D homology models. Thermal denaturation data validated experimentally this view (measured $T_m$ values of about 60 ˚C are for instance comparable to values measured for the HIV-1 capsid protein by similar means [47]), also experiments that showed that hetero-associations are reluctant to exchange their composing monomers. However, DSF and DLS data indicated that *Syn6803* K3/K4 and, more surprisingly, K1/K2 hetero-hexamers denature at lower temperatures than the homo-hexamers. The crystallization of CcmK4 from CcmK3/K4 samples might also point to a higher stability of homo-hexamers.

Understanding the reason of CcmK paralog multiplicity remains challenging. Globally, data collected on deletion mutants suggest that CcmK function might differ depending on the β-cyanobacteria, and possibly in relation with this, on environmental conditions. Redundant roles were initially proposed for *Syn7942* CcmK3 and CcmK4 to explain that photoautotrophic growth was compromised only with the Δ*ccmK3*-Δ*ccmK4* double mutant, but not with each individual knockout strain [48]. These observations were partly revoked recently by experiments that showed that the ccmK4 knockout grew 2.5 times slower than the WT strain [34], similarly to early observations on a Δ*ccmK4 Syn6803* strain [49]. These data would therefore point to CcmK3 and CcmK4 playing different roles. Though deletion of *Syn7942 ccmK3* gene was without phenotypic consequence, the systematic co-occurrence of *ccmK3*/*ccmK4* genes in 206 out of 227 β-cyanobacteria genomes [43] should be taken as indicative of a non-redundant role. That none of the two occurs alone strongly suggests that at least one of the functions of CcmK3 and CcmK4 must be different from each other and possibly complementary. The importance of CcmK3 could have been missed if it manifested only under certain environmental conditions that were not covered by previous studies. A precedent supports this possibility: the variable growth sensitivity of *Syn7942* to deletion of the *ccmK4* gene, depending on the culturing pH [34].

A shell-capping role was recently proposed for *Syn7942* K3/K4 hetero-hexamers [34]. In such a scenario, K3/K4 associations would position atop other hexamers embedded on the shell layer, presumably modifying their permeability. The interaction would be mediated by ionic residues (E98/R101) present in the C-terminal helix of *Syn7942* CcmK3. The proposition was based on the characterization by size-exclusion chromatography of K3/K4 species with the apparent size of a dodecamer (supposedly two stacked hexamers), and on the assumption that

K3 assembly must be hampered in virtue of the replacement of three residues that participate to inter-hexamer contacts with all other CcmK, and generally across BMC-H. Actually, AFM data presented here demonstrated a decline in assembly tendency of *Syn6803* K3/K4 and more surprisingly of the K1/K2 hetero-hexamers–residues implicated in hexamer contacts are identical in K1 and K2, indicating that the assembly defects are elicited differently (e.g. the different tendency of each paralog to form curved assemblies [23])–. Also agreeing with a peripheral role of K3/K4 complexes, the two YFP-labeled proteins were visualized in carboxysomes within *Syn7942* cells [50], whereas only CcmK4 was detected by MS when wild-type carboxysomes were purified [51]. This suggests that peripheral K3/K4, but not shell integrated K4 homo-hexamers, might have been lost during carboxysome isolation. Unfortunately, other known carboxysome components (e.g. CcmO, CcmN) were neither detected, making difficult to conclude on the real reasons behind the absence of CcmK3 detection. Investigating the fate of YFP-labeled K3 and K4 after purification of the corresponding fluorescent carboxysomes might permit to clarify this point. It is important to point out, however, that whilst *Syn6803* K1/K2 displayed the hexamer-dodecamer equilibrium characteristic of the K2 paralog, SEC-HPLC and native ESI-MS data proved that *Syn6803* K3/K4 behaves as hexamer in solution (fitting with a replacement of E98 by Ala in *Syn6803* K3). Although we cannot exclude that dodecamers formed under unexplored experimental conditions, our data presently disagree with a shell-capping function of *Syn6803* K3/K4.

In our opinion, a scenario with K3/K4 associations embedding within shells should not be fully ruled out. AFM data indicated that K3/K4 assemblies form, although giving rise to 2D networks of lower quality than those obtained with K4. The low K3:K4 stoichiometry and strong cooperativity of interactions established by shell components might still permit K3/K4 incorporation within shells. Indeed, AFM data proved that the mutation of one or even two of the key residues for inter-hexamer contacts does not always suffice to abolish BMC-H assembly [22, 23]. Similarly, R79A and N29A PduA mutants were found to accompany purified Pdu BMC, albeit with phenotypic features indicative of damaged shells [52]. If such scenario was right, local defects at contacts between K3 and neighboring hexamers might then constitute entry points for the exchange of subunits or the action of dedicated editing machineries (e.g. chaperones or proteases). Though not compulsory, the lower stability of K3/K4 hetero-hexamers might facilitate the remodeling/editing processes. In that manner, shell properties would be readapted to changes of environmental conditions in a simpler and less energy-consuming manner than if new compartments had to be built. Transcriptomic data indicated that the *Syn6803* K3-K4 operon is active under almost the full set of screened culturing conditions, whereas transcription of the K1/K2–containing operon seemed triggered by light [42]. It is therefore possible that the K3-K4 to K1-K2 ratio in *Syn6803* shells shifts depending on cellular age and/or environmental conditions, as suggest the increase of the number of K3 and K4 functional units measured for *Syn7942* carboxysomes, when comparing cells growing in the presence of 3% $CO_2$ or under atmospheric air conditions [50]. Carboxysome aggregation revealed in early stages of biogenesis and later migration within cells might also fit this model [53, 54], especially considering that *Syn7942* K3 (also K2) seem to interact with components that work to maintain a correct carboxysome distribution [55]. Intriguingly, carboxysome aggregation was also noticed inside a ΔCcmK3/K4 *Syn7942* double knockout cells [48].

CcmK3 might also serve to regulate the incorporation of other CcmK in shells. Thus, the expression of K3 in *E. coli* was found to alter the solubility of other co-expressed paralogs (S3 Fig), a possibility reaffirmed in experiments of co-expression of the four CcmK (S4 Fig). The effect was especially notorious on K1, a protein that was absent from soluble fractions in combinations with K3 (white arrows in S3B and S4B Figs). Accordingly, CcmK3 would act as a "precipitation trap" that would exhaust K1 arrival to shells, maybe allowing an enrichment by

the K4 paralog or preventing formation of K1/K2 hetero-hexamers. This last possibility agrees with the absence of purified material with cases 1 and 2 of S4 Fig. We indicate, however, that soluble K1-FLAG was still visible in combinations with K3-His$_4$ (black arrows in S3B Fig), although with smaller intensities than when co-expressed with other His$_4$-tagged proteins. The observation would be also irrelevant if the expression from K1-K2 and K3-K4 operons was decoupled in time in the natural host.

Although these studies were conducted in *E. coli*, the observation of K3/K4 compatibility between *Syn6803*, *Syn7942* and *Hal7418* paralogs strongly suggests that hetero-hexamers must occur in cyanobacteria too. Additional investigations are necessary to clarify this point, also to establish functional differences among species, to clarify paralog redundancies and to address more specifically hetero-hexamer function. With the exception of the mentioned study of the ΔCcmK4 strain, the physiological importance of individual *Syn6803* CcmK1, CcmK2 or CcmK3, or of combined CcmK1/K2 or CcmK3/K4 couples remains unexplored, to the best of our knowledge. Another point that would merit attention is the study of cross-interactions implying BMC-T too (also belonging to Pfam000936). Indeed, we collected preliminary data validating some associations with *Syn6803* CcmO and CcmP, and n-ESI-MS data were presented in a Ph.D. thesis, together with a likely over-simplified interpretation that needs to be reconsidered [44]. Future experiments are also necessary to investigate the connection between regulatory mechanisms and BMC-H promiscuity in species harboring several BMC, also to ascertain whether disrupted shells, which form in Eut-unregulated *Salmonella enterica*, could be caused by the potential integration in shells of heteromers combining monomers from different BMC types (and not of homo-hexamers from different BMC) [46].

Some published data will also need to be reinterpreted in the light of novel findings. For instance, Cai *et al.* presented data (basically fluorescence images) to support the formation of BMC chimeras integrating CsoS1 (i.e. homo-hexamers) from *Prochlorococcus marinus* str. MIT9313 into β-carboxysome shells from *Syn7942* [7]. Yet, the formation of hetero-hexamers combining CsoS1 and CcmK subunits should not be excluded. In fact, considering that the YFP domain attached to CsoS1 monomers is considerably bulkier than SUMO domains exploited by the same authors to prevent BMC-H assembly *in vitro* [56], the resulting CsoS1-YFP hexamers should also be assembly-incompetent. If this reasoning is correct, the fluorescence puncta observed *in vivo* should be the result of an incorporation of CsoS1-YFP/CcmK hetero-hexamers in the compartment shell.

## Supporting information

**S1 Table. Analysis of native-MS and MS-MS data collected on CcmK1/K2 and CcmK3/K4 hetero-hexamers.**
(DOCX)

**S2 Table. CcmK sequence identity scores.**
(TIF)

**S3 Table. Crystallographic data and refinement statistics.**
(DOCX)

**S4 Table. Differential Scanning Fluorimetry and Dynamic Light Scattering results.**
(DOCX)

**S1 List. Preparation of pET26b-based vectors for studies of CcmK coexpression.**
(DOCX)

**S1 Fig. Expression and solubility of untagged *Syn6803* CcmK paralogs in *Escherichia coli*.**
Left panel: total cellular contents, right: material remaining soluble in supernatants after lysis
and centrifugation at 20000 g. CcmK monomer bands are expected within the 10–15 kDa
range. "Empty" corresponds to cells transformed with the negative control pET15b vector
lacking the *ccmK* gene.
(TIF)

**S2 Fig. Expression and solubility of tagged *Syn6803* CcmK paralogs in *Escherichia coli*.**
Coomassie-stained SDS-PAGE views of cellular and soluble fractions prepared from BL21
(DE3) strains transformed with pET15b-based plasmids permitting expression of CcmK pro-
teins tagged with the indicated peptides. On top, total cellular expression levels are shown, bot-
tom part is for material remaining in supernatants after lysis and centrifugation at 20.000 g. *A*,
results collected for the N-ter tagged protein versions. *B*, similar data collected for C-ter tagged
proteins. Please notice that the relative vertical positioning of bands might differ slightly, as a
consequence of the image mounting process. Images from seven different gels needed to be
spliced and rearranged to prepare the figure.
(TIF)

**S3 Fig. Hetero-hexamer formation with combined *Syn6803* CcmK paralogs.** *A*, Schematic
representation of constructed expression vectors. Two ORFs coding for CcmK proteins tagged
at either N- or C-terminus with His4 and FLAG peptides were engineered between T7 pro-
moter and terminator sequences (blue and empty circles, respectively). Studied cases are listed
in the table. In bold and underlined are the combinations of the same paralog expected to pro-
duce doubly-tagged homo-hexamers (positive controls). Black letters are used for His4-carry-
ing subunit, violet for the FLAG-tagged proteins. *B*, Coomassie-stained SDS-PAGE showing
total cellular contents (C), soluble material remaining in supernatants after lysis and centrifu-
gation (S) and purified fractions (P). Only the portion of the gels presenting the region where
CcmK monomers appear is shown. The approximate position of the different co-expressed
partners is indicated on the right. Please notice that, as a consequence of partial proteolysis of
C-ter CcmK4 proteins, their emplacement cannot be unambiguously indicated. Besides, the
relative vertical positioning of bands in gels might be slightly erroneous, as a consequence of
image mounting process. White arrows are to indicate the absence of CcmK1-His$_4$ soluble
bands in combinations showing expression, whereas black arrows highlight CcmK1-FLAG
and CcmK2-FLAG solubility that seems unaffected by the presence of CcmK3. The figure
combines data from six different gels spliced and rearranged appropriately.
(TIF)

**S4 Fig. CcmK hetero-association occurrence upon co-expression of all *Syn6803* CcmK.** *A*,
Schematic representation of screened vectors. Four ORFs coding for CcmK proteins tagged at
either N- or C-terminus with His4 (black), Strep (blue), FLAG (violet) or HA (red) peptides
were engineered between T7 promoter and terminator sequences (blue and empty circles,
respectively). Studied combinations are listed in the table below. Whether asterisk is written
on before or after the paralog name denotes N- or C-tagging emplacement, respectively. *B*,
Coomassie-stained SDS-PAGE showing total cellular contents. (C), soluble material remaining
in supernatants after lysis and centrifugation (S) and purified fractions (P) prepared after cul-
turing BL21(DE3) transformed with plasmids schematized in panel A in auto-induction
media. Results are organized as in the table, depending on the position of the FLAG-tag with
regard to the protein placed on the third cassette (violet). Only the portion of the gels present-
ing the region where CcmK monomers appear is shown. White arrows indicate an absence of
CcmK1 soluble bands in combinations showing expression. *C*, Western blot analysis of

TALON-purified fractions. Detection on PVDC membranes was effected stepwise, a first incubation being performed with a mixture of mouse antiFLAG and mouse antiHA, followed by a second incubation with a mixture containing a secondary IgG antimouse-AP fusion plus streptactin-AP conjugate. The vertical position of protein bands in images B and C are not strictly matched with each other. The preparation of panels B and C required cropping and rearrangements from data collected in two gels, for each case. Please refer to M&M for further details.
(TIF)

**S5 Fig. Homology models of Syn*6803* CcmK3 and molecular dynamics.** *A*, Structures built by SWISS-MODEL (CcmK3_1, blue) and PHYRE2 (CcmK3_2, magenta), shown after alignment of the two colored monomers. For clarity, only the backbone trace is shown. Other neighbor monomers of the hexamer are represented as grey ribbons (only for CcmK3_1). Residues cumulating major differences are highlighted by thicker lines and include side-chain atoms. The two regions squared on the top view are zoomed and rotated in bottom images. The arrow indicates the position of residues lining the hypothetical pore, which would be clogged in CcmK3. *B*, Results of all-atom MD simulations run on CcmK3 homo-hexamer models and on CcmK3/K4 hetero-hexamers built after replacement of a single monomer of CcmK4 hexamer (PDB code 2A10) by either CcmK3_1 or CcmK3_2 models. RMSD deviations were measured for main-chain atoms of snapshot structures (recorded every 250 ps) compared to their position in the starting structure. Values estimated for the CcmK3_1 hexamer are shown in green, for CcmK3_2 hexamer in blue, CcmK3_1/CcmK4 hetero-hexamer in black and CcmK3_2/CcmK4 in red. Two independent 20 ns MD were run for each case, with analogous results.
(TIF)

**S6 Fig. Potential consequences on the structure and electrostatic properties of the formation of Syn*6803* CcmK3/K4 hetero-hexamers.** *A*, Comparison of structural details in different CcmK homo-hexamers, including the CcmK3_1 homology model built by SWISS-MODEL (CcmK3_1, blue), and the CcmK3_1/K4 hetero-hexamer model. All structures were pre-aligned to generate equivalent views. One of the monomers is shown in blue (corresponding to the CcmK3_1 subunit in the hetero-hexamer), neighbor ones are shown in grey ribbon representation. For clarity, only the backbone trace is shown. Residues that could contribute differently to inter-monomer contacts are colored as in [Fig 4](), with side-chain atoms drawn as balls-and-sticks. Arrows indicate the position of central hexamer pores, which is clogged in CcmK3. Ionic interactions are illustrated as red dashes. *B*, Comparison of surfaces of homohexamers and the CcmK3/K4 hetero-hexamer, colored according to electrostatic potential. Top and bottom images correspond to views of concave and convex sides, respectively. The approximate emplacement of the CcmK3 subunit in the CcmK3/K4 heterohexamer is indicated with dashed lines.
(TIF)

**S7 Fig. Hetero-hexamer formation between *Syn7942* CcmK3 and CcmK4 proteins.** Coomassie-stained SDS-PAGE showing total cellular contents (C), soluble material remaining in supernatants after lysis and centrifugation (S), and TALON-purified fractions (P). Left panel correspond to samples prepared from cellular cultures that were induced in IPTG (200 μM), whereas on the right are presented data from samples produced under auto-induction conditions. Top lines indicate the combination of isoforms co-expressed together, as well as the tagging identity. Tag emplacement at either N-ter or C-ter is indicated with an asterisk written before or after the paralog abbreviation, respectively. The figure combines data from four

different gels, which needed to be spliced and rearranged.
(TIF)

## Acknowledgments

We thank the scientific staff of the European Synchrotron Radiation Facility (Grenoble, France) and ALBA (Barcelona, Spain) for the use of their excellent data collection facilities. The DLS, DSF, and macromolecular crystallography equipment used in this study are part of the Integrated Screening Platform of Toulouse (PICT, IBiSA).

## Author Contributions

**Conceptualization:** Luis F. Garcia-Alles.

**Data curation:** Luis F. Garcia-Alles, Laurent Maveyraud, Renato Zenobi.

**Formal analysis:** Luis F. Garcia-Alles, Katharina Root, Laurent Maveyraud.

**Funding acquisition:** Lionel Mourey, Renato Zenobi, Gilles Truan.

**Investigation:** Luis F. Garcia-Alles, Laurent Maveyraud.

**Methodology:** Luis F. Garcia-Alles, Katharina Root, Laurent Maveyraud, Nathalie Aubry, Eric Lesniewska.

**Project administration:** Luis F. Garcia-Alles.

**Resources:** Laurent Maveyraud, Eric Lesniewska, Lionel Mourey, Renato Zenobi, Gilles Truan.

**Supervision:** Luis F. Garcia-Alles, Renato Zenobi.

**Validation:** Laurent Maveyraud.

**Visualization:** Luis F. Garcia-Alles, Laurent Maveyraud.

**Writing – original draft:** Luis F. Garcia-Alles.

**Writing – review & editing:** Laurent Maveyraud, Lionel Mourey, Renato Zenobi.

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
