## [Decision Letter · Decision Letter 0]

5 Sep 2019

[EXSCINDED]

PONE-D-19-22837

Occurrence and stability of hetero-hexamer associations formed by β-carboxysome CcmK shell components

PLOS ONE

Dear PhD Garcia-Alles,

Thank you for submitting your manuscript to PLOS ONE. After careful consideration, we feel that it has merit but does not fully meet PLOS ONE’s publication criteria as it currently stands. Therefore, we invite you to submit a revised version of the manuscript that addresses the points raised during the review process.

Both reviewers are enthusiastic about this manuscript for publishing on PLoS One. They only proposed minor concerns. Please refer to the individual comments and respond to them. Briefly, they suggest the authors revise the figures for a clearer presentation and make the discussion section more concise.  

We would appreciate receiving your revised manuscript by Oct 20 2019 11:59PM. To enhance the reproducibility of your results, we recommend that if applicable you deposit your laboratory protocols in protocols.io, where a protocol can be assigned its own identifier (DOI) such that it can be cited independently in the future. For instructions see: http://journals.plos.org/plosone/s/submission-guidelines#loc-laboratory-protocols

We look forward to receiving your revised manuscript.

Kind regards,

Zhicheng Dou, Ph.D.

Academic Editor

PLOS ONE

Journal Requirements:

'We thank the scientific staff of the European Synchrotron Radiation Facility (Grenoble, France) and ALBA (Barcelona, Spain) for the use of their excellent data collection facilities. The DLS, DSF, and

 macromolecular crystallography equipment used in this study are part of the Integrated Screening

 Platform of Toulouse (PICT, IBiSA). Financial support for the mass spectrometry studies by the Swiss National Science Foundation (grant number 200020_159929) is acknowledged. AFM work was funded

by FEDER grants.'

'LFGA, LM, NA, EL, LM and GT work received no specific funding for this work. KR and RZ acknowledge

financial support for mass spectrometry studies by the Swiss National Science Foundation (grant number 200020_159929). The funders had no role in study design, data collection and analysis, decision to publish, or preparation of the manuscript.'

Additional Editor Comments (if provided):

Both reviewers are enthusiastic about this manuscript for publishing on PLoS One. They only proposed minor concerns. Please refer to the individual comments and respond to them. Briefly, they suggest the authors revise the figures for a clearer presentation and make the discussion section more concise.

Reviewers' comments:

Reviewer's Responses to Questions

**Comments to the Author**

1. Is the manuscript technically sound, and do the data support the conclusions?

Reviewer #1: Yes

Reviewer #2: Yes

2. Has the statistical analysis been performed appropriately and rigorously? 

Reviewer #1: N/A

Reviewer #2: Yes

3. Have the authors made all data underlying the findings in their manuscript fully available?

Reviewer #1: Yes

Reviewer #2: Yes

4. Is the manuscript presented in an intelligible fashion and written in standard English?

Reviewer #1: Yes

Reviewer #2: No

5. Review Comments to the Author

Reviewer #1: This manuscript provides compelling evidence for the specific association of ccmK1 with ccmK2, and ccmK3 with ccmK4. These paralogous carboxysome shell proteins are located in different regions of cyanobacterial genomes--cmmK1 and 2 are associated with genes encoding other carboxysomal components, while ccmK3 and 4 are colocated elsewhere.

Expressing these shell proteins in a variety of combinations, and purifying the hexamers, showed K1/K2 associations and K3/K4 associations.

Frustratingly, K3 could not be co-crystalized with K4, but the other experiments do seem to show that these two proteins do indeed associate.

My suggestions are relatively minor and follow below:

Line 28: replace ‘permitted to localize’ with ‘permitted the localization of’

Line 29-30: ‘Attempts to crystallize these CcmK3/K4 associations conducted to the unambiguous elucidation of a CcmK4 homo-hexamer structure’ needs to be rephrased

Line 32: ‘in frame with’ should be replaced with ‘within the context of’

Line 66: ‘single protein paralogs’ Would rephrase as ‘single proteins’

Lines 95 and 96, and throughout the manuscript: double-check to be certain that taxon names are italicized

Line 107: ‘chimio’ should be replaced with ‘chemically’

Line 135: delete ‘for ulterior analyses’

Line 136: delete ‘steps’ after ‘3-4’

Line 143: replace ‘ulterior’ with ‘subsequent’

Line 163: delete ‘next’

Line 202: remove ‘equipment’

Line 210: remove ‘a’

Line 264: ‘permitting the production of soluble CcmK1 (K1) and CcmK2 (K2) with any of the 4 peptides’. I do not follow this.

Line 292: ‘Such bands were net for K1, K2 and K4 cases’…? Clarify?

Line 386: replace ‘permitted to identify’ with ‘identified’

Fig. 3: The information schematized in Fig. S4A should probably be placed somehow in figure 3 to help the reader follow this

Fig 5C—only see one band. Should we see two, since the text is suggesting K3K4 interaction? Perhaps put more info in the text or figure legend to understand what we are seeing here?

Line 463: ‘Besides’ should probably be replaced with ‘Furthermore’

Line 482: Replace ‘as indicate the WB’ with ‘as indicated by the WB’

Line 508: Replace ‘revealed’ with ‘was’

Line 514: Replace ‘to monitor’ with ‘the monitoring of’

Fig. 6: use larger symbols so the traces can be distinguished

Line 546: insert ‘the opportunity’ after ‘permit’

Line 575: replace ‘non-rare’ with ‘many’

Reviewer #2: Garcia Alles et al. describe a series of experiments designed to elucidate interactions between different CcmK* homologs of Synechocystis. To their credit, the work is very thorough, with investigations of the influence of different tags and tag positions on subunit interactions, and experiments where all 4 proteins are simultaneously expressed. The initial conclusions from Western blotting are backed by mass spectrometry analysis. The CcmK1/CcmK2 complex and CcmK3/CcmK4 complex are also tested for their ability to pack laterally using AFM. Homology modelling and sequence comparisons are used to investigate the CcmK3/CcmK4 complex. The stability of this complex and ability of subunits to exchange is also tested. In general, this is a careful, very thorough study; the authors were unfortunate in that their major conclusion was anticipated by a competitor’s work, but this paper adds depth and nuance, and certainly deserves a place in PLOS One.

While the work seems technically sound, I do think the presentation could be improved. Phrasing is often awkward, sometimes to the point where the author’s intentions are not clear. The paper would greatly benefit from editorial attention from a native english speaker. The figures are also difficult to follow in terms of the experimental design and what a positive/negative readout tells you. I think the authors should consider adding some of the explanatory panels currently in the supplementary figures to the main figures. Fig. 3 in particular would benefit.

The discussion is overly long (seven pages) and rambling. It reiterates many of the points made in the introduction and results at some length. It also introduces a great deal of fairly speculative ideas. I think the paper would be much more readable if the authors tried to limit this, focussing on the ideas that they think most likely, or a handful of their best supported speculative ideas.

The section on sequence and structural arguments for CcmK3 forming a complex with CcmK4 is based largely on arguments from homology models and molecular dynamics, which have fairly strong inherent biases towards the starting model. I feel this section could again be shortened.

Figure 1 – I am not opposed to the idea of using lines to guide the eye, but perhaps the vertical lines could be less prominent where a single gel is used, in order to not obscure where adjacent lanes are from the same gel. E.g. a less thick, and perhaps dashed line would serve equally well for demarcation.

637 – lack of evidence from coevolutionary algorithms may simply reflect that there are not enough distinct beta-cyanobacterial sequences.

The legend for Fig 3 does not really allow the experiment to be understood as presented. For example, on what basis is each lane labelled with the black and purple letters at the top? What is the difference between lanes? Are all tags actually present in each experiment? I understand the conclusion the authors reach, and can see evidence for individual interactions, but don’t understand how each experiment was set up, or how to interpret the absence of WB signal with each of the tags being interrogated.

Fig 4B again lacks context. What model is this? One of the homology models, or a post MD model? The orientation of the image is also not immediately clear to the reader. Picking standard orientation (concave or convex surface) and perhaps using surface colour to show how these residues protrude might help.

Fig S3 – what does underlining indicate in the table?

6. PLOS authors have the option to publish the peer review history of their article (what does this mean?). If published, this will include your full peer review and any attached files.

Reviewer #1: No

Reviewer #2: No

---

## [Author Response · Author response to Decision Letter 0]

16 Sep 2019

Please find below in black letters our reply to reviewers comments. Other minor changes were introduced, mainly with the intention to improve the discussion. Most important are mentioned in the last portion of this letter, others can be easily visualized in the “Revised Manuscript with Track Changes” version.

Reviewer #1: 

This manuscript provides compelling evidence for the specific association of ccmK1 with ccmK2, and ccmK3 with ccmK4. These paralogous carboxysome shell proteins are located in different regions of cyanobacterial genomes--cmmK1 and 2 are associated with genes encoding other carboxysomal components, while ccmK3 and 4 are colocated elsewhere.

Expressing these shell proteins in a variety of combinations, and purifying the hexamers, showed K1/K2 associations and K3/K4 associations.

Frustratingly, K3 could not be co-crystalized with K4, but the other experiments do seem to show that these two proteins do indeed associate.

My suggestions are relatively minor and follow below:

Line 28: replace ‘permitted to localize’ with ‘permitted the localization of’

It was replaced

Line 29-30: ‘Attempts to crystallize these CcmK3/K4 associations conducted to the unambiguous elucidation of a CcmK4 homo-hexamer structure’ needs to be rephrased

Replaced by “The crystallization of these CcmK3/K4 associations conducted to the elucidation of a structure corresponding to the CcmK4 homo-hexamer”. We hope it will be better.

Line 32: ‘in frame with’ should be replaced with ‘within the context of’

Line 66: ‘single protein paralogs’ Would rephrase as ‘single proteins’

Lines 95 and 96, and throughout the manuscript: double-check to be certain that taxon names are italicized

Line 107: ‘chimio’ should be replaced with ‘chemically’

Line 135: delete ‘for ulterior analyses’

Line 136: delete ‘steps’ after ‘3-4’

Line 143: replace ‘ulterior’ with ‘subsequent’

Line 163: delete ‘next’

Line 202: remove ‘equipment’

Line 210: remove ‘a’

All these proposed changes have been introduced.

Line 264: ‘permitting the production of soluble CcmK1 (K1) and CcmK2 (K2) with any of the 4 peptides’. I do not follow this.

We replaced the sentence by: “Thus, CcmK1 (K1) and CcmK2 (K2) were expressed and remained soluble regardless of the peptide tag identity at C-terminus (Fig. S2)” that we hope will be clearer. 

Line 292: ‘Such bands were net for K1, K2 and K4 cases’…? Clarify?

We introduced two modifications: 1) we specified in legend to Fig. S3A the meaning of underlined combinations shown in the table, something that we forgot to mention in the submitted version; 2) in the main text now, it is written: “Cases combining His4- and FLAG-tagged versions of the same paralog served as positive controls (underlined in table of Fig. S3A), providing an indication of signal level attained for homo-hexamers. Such bands were clearly detected for K1-His4/K1-FLAG, K2-His4/K2-FLAG, K4-His4/K4-FLAG and His4-K4 /K4-FLAG.” 

Line 386: replace ‘permitted to identify’ with ‘identified’

We did. 

Fig. 3: The information schematized in Fig. S4A should probably be placed somehow in figure 3 to help the reader follow this

Figure 3 has been modified accordingly. Now, two panels are present and numerical correspondences between gene constructs and analyzed cases are given for better clarity. We hope it is now better and we apologize for complicating the reading of the first submitted version of the manuscript. 

Fig 5C—only see one band. Should we see two, since the text is suggesting K3K4 interaction? Perhaps put more info in the text or figure legend to understand what we are seeing here?

Unfortunately, the two proteins present an almost identical electrophoretic mobility. This is now mentioned in the figure legend: “Please notice that His4-CcmK4 and CcmK3-FLAG bands could not be resolved.”

Line 463: ‘Besides’ should probably be replaced with ‘Furthermore’

Line 482: Replace ‘as indicate the WB’ with ‘as indicated by the WB’

Line 508: Replace ‘revealed’ with ‘was’

Line 514: Replace ‘to monitor’ with ‘the monitoring of’

All proposed changes have been introduced.

Fig. 6: use larger symbols so the traces can be distinguished

The figure has been modified accordingly. However, larger symbols imposed to show alternate data points (1 every 2), otherwise it became impossible to recognize each trace. This is mentioned in the legend.

We hope the modifications are helpful.

Line 546: insert ‘the opportunity’ after ‘permit’

Line 575: replace ‘non-rare’ with ‘many’

These changes were introduced in the revised version.

We sincerely thank the reviewer for helping us in ameliorating the readability of our study.

Reviewer #2: Garcia Alles et al. describe a series of experiments designed to elucidate interactions between different CcmK* homologs of Synechocystis. To their credit, the work is very thorough, with investigations of the influence of different tags and tag positions on subunit interactions, and experiments where all 4 proteins are simultaneously expressed. The initial conclusions from Western blotting are backed by mass spectrometry analysis. The CcmK1/CcmK2 complex and CcmK3/CcmK4 complex are also tested for their ability to pack laterally using AFM. Homology modelling and sequence comparisons are used to investigate the CcmK3/CcmK4 complex. The stability of this complex and ability of subunits to exchange is also tested. In general, this is a careful, very thorough study; the authors were unfortunate in that their major conclusion was anticipated by a competitor’s work, but this paper adds depth and nuance, and certainly deserves a place in PLOS One.

While the work seems technically sound, I do think the presentation could be improved. Phrasing is often awkward, sometimes to the point where the author’s intentions are not clear. The paper would greatly benefit from editorial attention from a native english speaker. The figures are also difficult to follow in terms of the experimental design and what a positive/negative readout tells you. I think the authors should consider adding some of the explanatory panels currently in the supplementary figures to the main figures. Fig. 3 in particular would benefit.

The reviewer is right. As indicated for reviewer 1, Figure 3 has now been modified. An explanatory panel was added and numerical correspondences between gene constructs and analyzed cases are given for better clarity. 

Similarly, we modified figure S4 to make things clearer. The same table with numeric correspondences is presented again, and the panel B was split in two pieces to permit a clearer attribution of data collected for each analyzed case 1 to 8.

Similarly, the table in Fig. S3 and labeling of Fig. 7 have been modified with the intention to make attribution of studied combinations to corresponding data clearer.

We hope the manuscript is now easier to follow. We apologize for complicating unnecessarily the reading of the first submitted version of the manuscript.

The discussion is overly long (seven pages) and rambling. It reiterates many of the points made in the introduction and results at some length. It also introduces a great deal of fairly speculative ideas. I think the paper would be much more readable if the authors tried to limit this, focussing on the ideas that they think most likely, or a handful of their best supported speculative ideas.

We agree the discussion is long. We tried to rearrange some paragraphs and to delete unnecessary portions. However, it was necessary to update the discussion in the light of very recently published data that demonstrate the presence of CcmK3 and CcmK4 in carboxysome shells (two new citations were introduced). In consequence, the text shrank only slightly, passing from 17715 characters to 15190 characters now (15% shorter).

I admit the discussion is still long, but I think it is necessary. We dedicate the first 2,5 pages to summarize and put together the new observations. The next 1,5 pages are necessary to recapitulate published data on CcmK3/CcmK4 knockouts, and to discuss our data within the frame of the proposed shell-capping scenario, which can be considered speculative. One page more is written to discuss two other possible functions that emerge on the light of our data, which we believe deserve consideration, in spite of the fact that they are speculative too. The last page intends to present perspectives for future experimentation.

We hope the reviewer will find the new version clearer, less repetitive.

The section on sequence and structural arguments for CcmK3 forming a complex with CcmK4 is based largely on arguments from homology models and molecular dynamics, which have fairly strong inherent biases towards the starting model. I feel this section could again be shortened.

We agree on the need to be cautious when considering these data, and that is why we intentionally wrote the paragraph in a conditional style. However, we think that 3D homology models and MD simulations still provide some clues on why CcmK3 might be more stable when embedded within CcmK4 monomers than alone. This could be of help for other investigations, for instance for crystallography attempts, or to compare the solution behavior of CcmK3 from different organisms.

Figure 1 – I am not opposed to the idea of using lines to guide the eye, but perhaps the vertical lines could be less prominent where a single gel is used, in order to not obscure where adjacent lanes are from the same gel. E.g. a less thick, and perhaps dashed line would serve equally well for demarcation.

White vertical lines have been introduced in Fig. 1 to highlight gel discontinuities in the mounted figure. We hope this is what reviewer asked for.

637 – lack of evidence from coevolutionary algorithms may simply reflect that there are not enough distinct beta-cyanobacterial sequences.

The reviewer is correct. We deleted in the revised version the sentence, to avoid unnecessary speculation.

The legend for Fig 3 does not really allow the experiment to be understood as presented. For example, on what basis is each lane labelled with the black and purple letters at the top? What is the difference between lanes? Are all tags actually present in each experiment? I understand the conclusion the authors reach, and can see evidence for individual interactions, but don’t understand how each experiment was set up, or how to interpret the absence of WB signal with each of the tags being interrogated.

As indicated above, Figure 3 has now been modified accordingly. An explanatory panel was added and numerical correspondences between gene constructs and analyzed cases are given for better clarity.

We hope it is now clearer. We apologize for the inconvenience while reading the first submitted version of the manuscript.

Fig 4B again lacks context. What model is this? One of the homology models, or a post MD model? The orientation of the image is also not immediately clear to the reader. Picking standard orientation (concave or convex surface) and perhaps using surface colour to show how these residues protrude might help.

In original Figure 4B legend was written: “Presumed localization on the Syn6803 CcmK3_1 modeled structure of some notable residue substitutions”. The CcmK3_1 model is the one prepared with SWISS-MODEL, as indicated in the main text, 415-429 : “To further visualize these differences, homology models were built using SWISS-MODEL (CcmK3_1) and PHYRE2 (CcmK3_2) algorithms. The resulting monomer models were similar…”. We therefore do not understand very the first question. However, to further clarify the text, we have replaced the legend sentence by “Presumed localization of some notable residue substitutions in the CcmK3 Syn6803 structure furnished by SWISS-MODEL (CcmK3_1 in the main text), when compared to other CcmK”

As far as the orientation problem and surface colors is concerned, we agree with the reviewer the images were not clear enough. We modified the two panels of the figure, and the corresponding legend. We hope things are now more easy to follow. 

Fig S3 – what does underlining indicate in the table?

 We apologize, we forgot to mention it. The legend now reads: “Studied cases are listed in the table. In bold and underlined are the combinations of the same paralog expected to produce doubly-tagged homo-hexamers (positive controls)”. 

We thank the reviewer for calling our attention on this detail.

---

## [Decision Letter · Decision Letter 1]

2 Oct 2019

Occurrence and stability of hetero-hexamer associations formed by β-carboxysome CcmK shell components

PONE-D-19-22837R1

Dear Dr. Garcia-Alles,

We are pleased to inform you that your manuscript has been judged scientifically suitable for publication and will be formally accepted for publication once it complies with all outstanding technical requirements.

With kind regards,

Zhicheng Dou, Ph.D.

Academic Editor

PLOS ONE

Additional Editor Comments (optional):

Both reviewers have a few minor suggestions to the revised manuscript. Please address these minor comments. In addition, please add additional explanations to the Figure 2 legend since the authors identified mislabeled annotation to the mass spectrometry data in this figure after the reviewers read the revised manuscript. 

Reviewers' comments:

Reviewer's Responses to Questions

**Comments to the Author**

1. If the authors have adequately addressed your comments raised in a previous round of review and you feel that this manuscript is now acceptable for publication, you may indicate that here to bypass the “Comments to the Author” section, enter your conflict of interest statement in the “Confidential to Editor” section, and submit your "Accept" recommendation.

Reviewer #1: All comments have been addressed

Reviewer #2: All comments have been addressed

2. Is the manuscript technically sound, and do the data support the conclusions?

Reviewer #1: Yes

Reviewer #2: Yes

3. Has the statistical analysis been performed appropriately and rigorously? 

Reviewer #1: N/A

Reviewer #2: Yes

4. Have the authors made all data underlying the findings in their manuscript fully available?

Reviewer #1: Yes

Reviewer #2: Yes

5. Is the manuscript presented in an intelligible fashion and written in standard English?

Reviewer #1: Yes

Reviewer #2: Yes

6. Review Comments to the Author

Reviewer #1: Thank you for addressing my comments. The diagram added to figure 3 really clarifies things, as does adding a statement for figure 5C that K3 and K4 cannot be distinguised based on electrophoretic mobility

Reviewer #2: The fairly minor comments I had have largely been dealt with. The discussion still feels overly long to me, but this ultimately is a matter of taste. I also feel that while the writing is improved, it is still a bit challenging to read in places.

A couple of minor points I noticed in passing:

Homologies should not be used in a quantitative context in a general rule as it simply means “related by common descent”. It is better to refer to this as sequence similarity or sequence identity, the phenomenon you can actually observe. E.g. line 567.

Abstract L 29 – Maybe: Attempts to crystallize a CcmK3/K4 complex resulted in only a structure of CcmK4 alone.

7. PLOS authors have the option to publish the peer review history of their article (what does this mean?). If published, this will include your full peer review and any attached files.

Reviewer #1: Yes: Kathleen M Scott

Reviewer #2: No

---

## [Editor Report · Acceptance letter]

4 Oct 2019

PONE-D-19-22837R1 

Occurrence and stability of hetero-hexamer associations formed by β-carboxysome CcmK shell components 

Dear Dr. Garcia-Alles:

I am pleased to inform you that your manuscript has been deemed suitable for publication in PLOS ONE. Congratulations! Your manuscript is now with our production department. 

With kind regards,

on behalf of

Dr. Zhicheng Dou 

Academic Editor

PLOS ONE